# Linking structural and functional changes during aging using multilayer brain network analysis
Gwendolyn Jauny[1], Mite Mijalkov[2], Anna Canal-Garcia [2], Giovanni Volpe [3], Joana Pereira [2,4], Francis Eustache [1] & Thomas Hinault [1] ✉

Brain structure and function are intimately linked, however this association remains poorly understood and the complexity of this relationship has remained understudied. Healthy aging is characterised by heterogenous levels of structural integrity changes that influence functional network dynamics. Here, we use the multilayer brain network analysis on structural (diffusion weighted imaging) and functional (magnetoencephalography) data from the Cam-CAN database. We found that the level of similarity of connectivity patterns between brain structure and function in the parietal and temporal regions (alpha frequency band) is associated with cognitive performance in healthy older individuals. These results highlight the impact of structural connectivity changes on the reorganisation of functional connectivity associated with the preservation of cognitive function, and provide a mechanistic understanding of the concepts of brain maintenance and compensation with aging. Investigation of the link between structure and function could thus represent a new marker of individual variability, and of pathological changes.

The brain is one of the most complex biological systems. One of its fascinating aspects, which remains largely unknown, is how wide varieties of brain rhythms and temporally-specific activity patterns[1] can emerge from a static network architecture[2]. Addressing this issue is a major fundamental endeavour for cognitive neuroscience, which can also improve our understanding of brain changes across the lifespan and our ability to detect pathological processes. Previous work has mostly focused on characterising brain structure (i.e., grey matter and white matter) or brain function (i.e., memory, motor function or cognitive control)[3]. These unimodal studies greatly advanced our understating of brain networks and of their associations with cognition[4]. However, brain network analysis methods, such as graph theory, have been applied across modalities to study the interaction between structure and function, showing strong associations between these dimensions[5,6]. Since these seminal studies, the relationship between brain structure and function has been the focus of intense reflection and methodological development since this relationship is central to many cognitive domains, evolves with age and is affected by pathologies[5]. Here, we investigate these issues in light of age-related brain changes associated with changes in brain structure that influence neural dynamics[7], which could further our understanding of the large heterogeneity of individual

cognitive trajectories observed during this life period. In particular, structure–function interactions could be central to further understanding the preservation (i.e., maintenance[8] or compensation[9]) or the decline of cognitive performance during ageing.

Studying the relationships between white matter fibres (acquired by DWI—diffusion-weighted imaging) and blood-oxygen-level-dependent (BOLD) signal (acquired by fMRI—functional magnetic resonance imaging), previous studies have shown correlations between brain structure and function throughout the lifespan, and particularly across development[10,11], and during the performance of cognitive tasks[12]. Also, in a healthy older population, Burzynska et al.[13] showed that individuals with preserved white matter fibre integrity had a higher BOLD signal associated with better cognitive performance (see also[14,15]). Many studies have thus focused on this link between structure and function using high-spatial-resolution techniques such as fMRI. However, due to their constrained temporal resolution, age-related changes in the dynamics of the involved networks remain largely understudied.

Previous work has also demonstrated interactions between brain structure and function using high-temporal resolution techniques, such as magnetoencephalography (MEG) or electroencephalography (EEG).

[1]Normandie Univ, UNICAEN, PSL Université Paris, EPHE, Inserm, U1077, CHU de Caen, Centre Cyceron, Neuropsychologie et Imagerie de la Mémoire Humaine, 14000 Caen, France. [2]Department of Clinical Neuroscience, Karolinska Institutet, Stockholm, Sweden. [3]Department of Physics, Goteborg University, Goteborg, Sweden. [4]Clinical Memory Research Unit, Department of Clinical Sciences, Lund University, Malmö, Sweden. ✉e-mail: thomas.hinault@inserm.fr

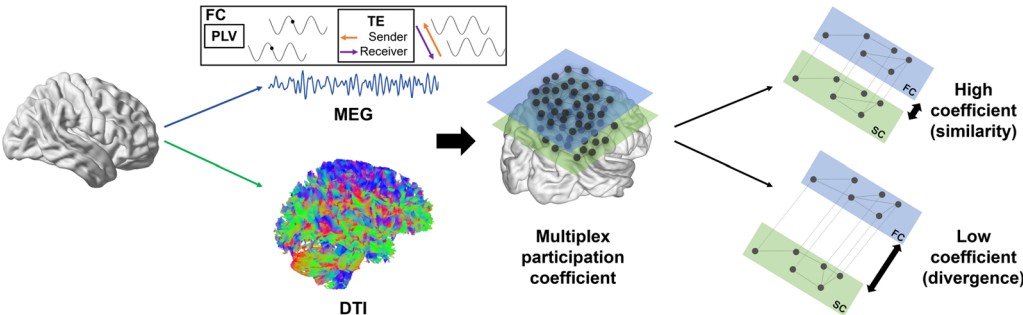

**Fig. 1 | Overview of the creation of the multiplex network from MEG and DWI data.** This multiplex network was built with two layers: one representing functional connectivity (FC) from MEG data, either PLV or TE data; the other layer representing structural connectivity (SC) from DWI (anisotropic fraction) data, i.e., FA data. MEG magnetoencephalography, DWI diffusion-weighted imaging, PLV phase locking value, TE transfer entropy, FC functional connectivity, SC structural connectivity.

Indeed, fluctuations in the synchrony and transfer entropy (i.e., directionality) of brain activity have long been considered as noise to be controlled, whereas today, they have been reappraised as a fundamental aspect of brain communication[16,17]. These studies have notably highlighted that EEG connectivity is associated with structural connectivity measures in young adults[18]. With healthy ageing, Hinault et al.[19,20] showed that a decrease in white matter fibre integrity is negatively associated with neural synchronisation between brain regions. However, for all these studies, the interpretation of these interactions is limited as it does not account for the full complexity of such a relationship.

A recent approach enables evaluating the relationships between different neuroimaging modalities by constructing a multiplex network model of the brain[21]. This approach allows the creation of a network in which each region is connected to itself across different layers[22]. This technique has already been used in pathology, such as schizophrenia[23] and Alzheimer's disease[24,25], allowing to highlight brain changes that were not detected in unimodal analyses. Recently, the study by Battiston et al.[26], investigating network connectivity by combining fMRI and DWI data in a two-layer multiplex network, revealed relevant relationships between structural and functional brain networks, showing that this technique is an appropriate choice for the study of brain network connectivity. Some studies have also investigated changes in functional connectivity in healthy[27] or pathological[28,29] participants using a multiplex approach applied on MEG data. Thus, multiplex brain networks can be used to study the structure-function relationship in healthy ageing. It seems, therefore, necessary to quantify the heterogeneity of this structure/function relationship in relation to cognitive heterogeneity. Moreover, previous work[30] suggested that alterations in brain structure in the presence of delayed and/or noisier brain communications. Such a combination of DWI (structural) and MEG (functional) data in a multiplex connectome in healthy ageing is therefore important to identify markers of individual differences and early brain ageing effects, preceding major structural changes and loss of functional communications. These changes can lead to deleterious functional consequences[20,31] or compensatory functional adjustments[32]. In particular, the functional role of the regions could be affected by changes in brain structure. Indeed, recent work using fMRI has shown in humans the presence of a functional asymmetry between brain regions in terms of afferent and efferent information transfer[33]. Other computer modelling work has also shown a relationship between network topology and information directionality, in particular, by identifying certain brain regions (or nodes) as targets and sources of information[34]. This method, therefore, appears ideal to clarify the association between brain structure and cortical dynamics, to identify the mechanisms underlying cognitive heterogeneity with ageing, and the functional adjustments allowing the maintenance of cognitive function.

Here, we propose a multiplex network approach with MEG and DWI data in the context of healthy ageing and the associated non-lesional brain changes[35] (see Fig. 1). We used the multiplex participation coefficient as an indicator of the similarity of connectivity between brain structure and function: a high level of this coefficient corresponded to a similarity of connectivity patterns between these modalities whereas a low level corresponded to a divergence of connectivity patterns between these modalities. We investigated age-related changes in brain structure and function in young and older healthy participants from the Cam-CAN database (Cambridge Centre for Aging and Neuroscience[36,37]). This database includes multimodal neuroimaging data (MEG, MRI and DWI) as well as cognitive performance evaluation for each individual. Our objectives were twofold: (i) To investigate changes in the interaction between structural integrity levels and synchronised functional networks between young and old individuals, with the underlying hypothesis that a decrease in white matter integrity could negatively impact brain function. (ii) To study the impact of such structure–function relationship on participants' cognitive performance, where we expected that these changes would be associated with cognitive performance and reveal unique individual differences therein. Compensatory adjustments or maintenance of brain function at the same level as young adults would result in the preservation of cognitive performance. Such results could clarify and better characterise maladaptive and compensatory brain communication changes in the presence of ageing structural networks.

## Results

Two groups of participants (22–29 years for the younger group and 60–69 years for the older group) were formed from the Cam-CAN[36,37] database. Connectivity analyses were performed on MEG data, and in particular, two measures were studied: phase locking value (PLV), which measures synchrony between regions, and transfer entropy (TE), which measures the directionality of the coupling between brain regions. TE is complementary to the synchrony measure (PLV) as it provides an estimation of directed connectivity. These measures have already been used in previous works investigating age-related changes[28,29]. The data from these two measures were combined with DWI data to form two multiplex structure–function networks (see Fig. 1). From these networks, the multiplex participation coefficient could be calculated. This coefficient was then studied to determine the level of similarity of connectivity between the two layers (structural and functional) of the network. The different phases of data processing, creation of multiplex networks and statistical analysis are described in the materials and methods section.

### The positive association between multiplex participation coefficients and cognitive performance in older adults

We first studied a multiplex network composed of PLV and DTI data. Our main objective was to study the effect of healthy ageing on structural and functional connectivity and its association with cognitive abilities (measured with neuropsychological tests assessing working and short-term memory, reasoning ability, executive functions and general cognitive functions; see materials and methods for more information). Thus, we determined which region and which frequency bands age-related changes in multiplex

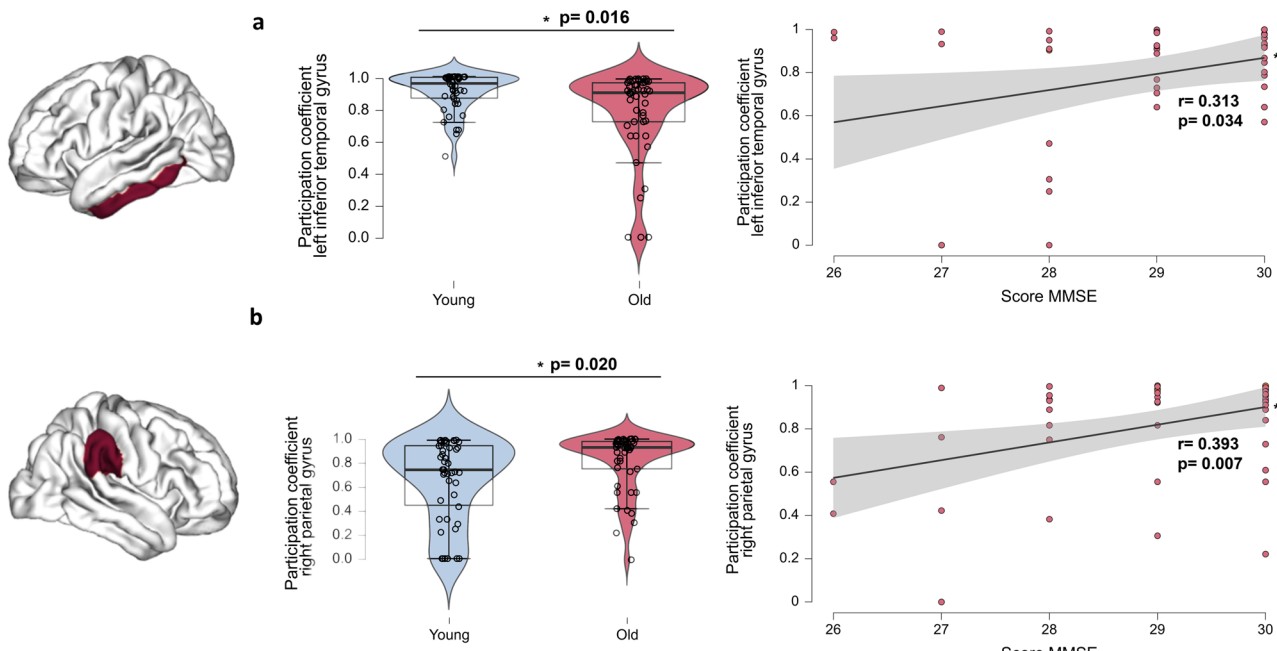

**Fig. 2 | Multiplex participation coefficient level differences between young and old groups and association with cognition. a** Distribution of the young and old groups in the left inferior temporal region (*t*-test) for the multiplex participation coefficient in the alpha frequency band for the measure of synchrony (PLV) and positive association between this level of multiplex participation coefficient and MMSE score. **b** Distribution of the young and old groups in the right parietal region (*t*-test) for the multiplex participation coefficient in the alpha frequency band for the measure of synchrony (PLV) and positive association between this level of multiplex participation coefficient and MMSE score in older adults. The level of education was controlled as a covariate. All results were adjusted for multiple comparisons using FDR corrections at *q* < 0.05. *n* = 46 participants per group. The black vertical line represents the standard error of the mean. *\*p* < 0.05 \*\**p* < 0.01.

participation coefficient could be associated with cognitive performance. First, we identified the regions and frequency bands that differed between age groups and were associated with cognition: the left temporal and right parietal regions in the alpha frequency band (these two regions showed, respectively, a decrease or an increase in participation in the older individuals compared to the younger). For other regions and frequency bands showing differences, they were not associated with cognitive performance, see Fig. S1 in supplementary. We found that, for both of these regions, increased multiplex participation coefficient levels were positively associated with cognitive performance in older adults (left temporal/MMSE test, *r* = 0.313, *p* = 0.034; right parietal/MMSE test, *r* = 0.393, *p* = 0.007; Fig. 2). No association was found in young adults.

**Maintaining a lower level of multiplex participation coefficient than younger adults is positive for the older population**
To further analyse these results, subgroup analyses were performed for these two regions. To do this, participants were grouped according to the level of participation coefficient in each region, forming two groups of older individuals. The older subgroups (i.e., Low participation, High participation; see Table S1 to Table S4 in supplementary data for the characteristics of each subgroup) did not differ on any measure (e.g., age, gender ratio, level of education and general cognitive performance) other than the level of multiplex participation coefficient (left temporal and right parietal regions). For the left temporal region, young adults differ from both older subgroups, and both subgroups also significantly differ from each other: the level of the participation coefficient was significantly higher for the High participation subgroup than the younger group (*p* = 0.009). The Low participation subgroup showed lower multiplex participation levels than both younger individuals and the High participation subgroup (*p* < 0.001 for both comparisons). The Low participation subgroup showed better cognitive performance on the VSTM test than the High participation subgroup (*r* = 0.584, *p* = 0.009; Fig. 3a). For the right parietal region, young adults differ from the High participation subgroup, but not with the Low participation subgroup. We observed that the Low participation subgroup (with

similar low participation as younger individuals, *p* = 0.962) showed better cognitive performance on the VSTM test (*r* = 0.475, *p* = 0.040; no association with cognition for the high participation older subgroup; Fig. 3b). Replication of the main findings using wPLI ((weighted Phase Lag Index analyses) see Fig. S2) (Fig. 3). Multiplex participation coefficient level differences between young and older subgroups and association with cognition.

**Age-related changes in network coupling directionality are positively associated with cognitive performance**
Following these results, we examined ageing effects and individual differences in these regions using directed functional couplings. For this, we used the multiplex network formed from TE and DTI data. For the right parietal region only, in the alpha band, we observed an increase in inward directionality (i.e., directed towards the right parietal region) in older individuals compared to younger individuals (*t*-test, *p* = 0.038; Fig. 4a). See Supplementary Fig. S3 for consistent results involving gamma frequency bands. This increased participation in the inward direction for the right parietal region with ageing was positively associated with performance in the VSTM test (*r* = 0.314, *p* = 0.034; Fig. 4c).

To further analyse these results, we investigated differences in the same subgroups as in the first part (PLV/DWI) of the results.

We observed that the Low participation subgroup, showing increased inward-directed couplings in the right parietal region, also showed better cognitive performance on the VSTM test (*r* = 0.463, *p* = 0.046; Fig. 3c) than the High participation subgroup. Supporting these results, the High participation older subgroup showed lower cognitive performance on the VSTM test (*r* = −0.491, *p* = 0.033; Fig. S3 in Supplementary data).

**The respective contribution of each network layer in younger and older adults**
Degree analyses (number of connections) were performed on the respective contribution of each layer and suggest that the structural layer makes the largest contribution to the reported results, as the degree was larger in the

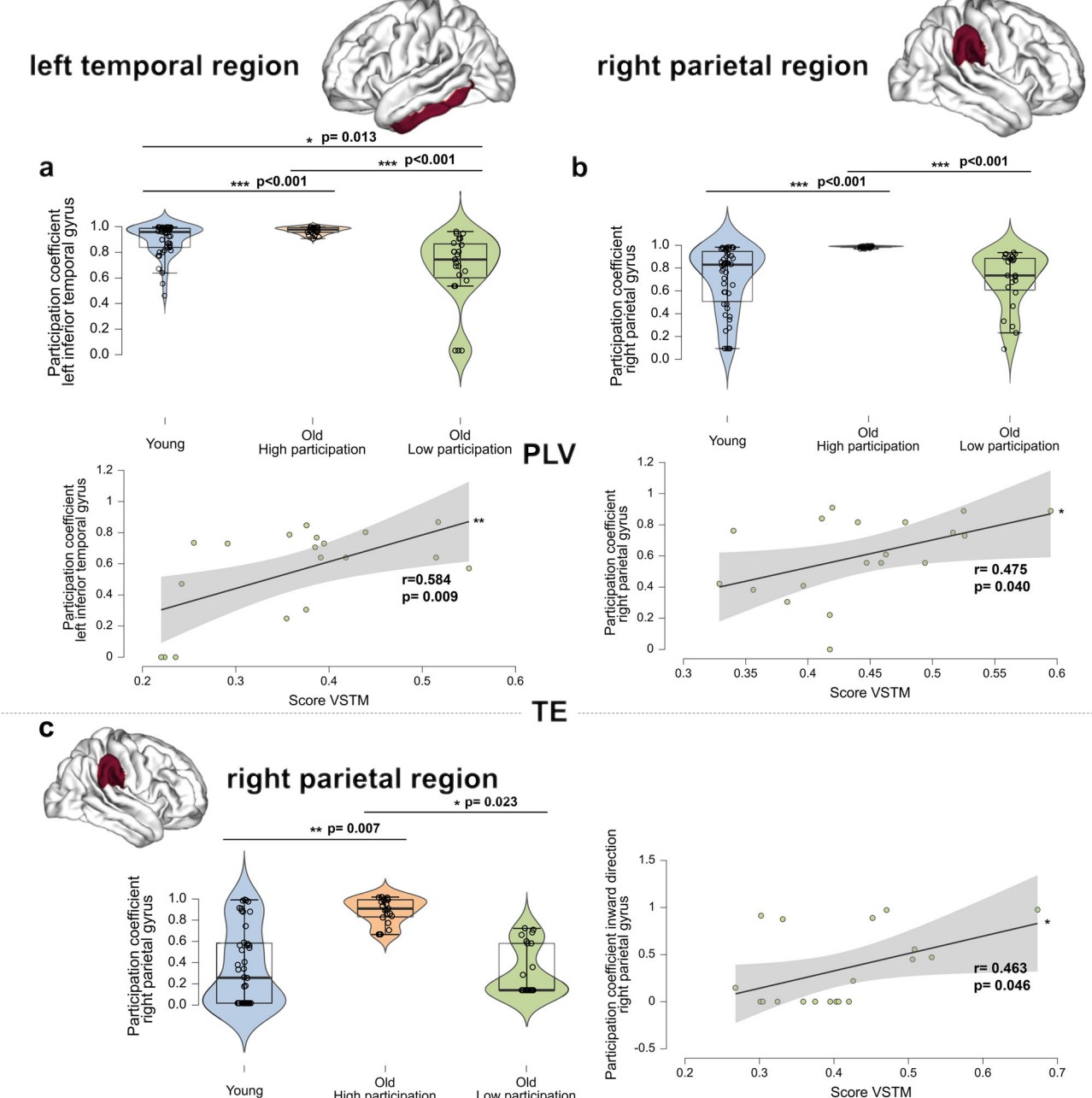

**Fig. 3 | Multiplex participation coefficient level differences between young and older subgroups and association with cognition. a** Distribution of young adults and older adults' subgroups for the multiplex participation coefficient in the left temporal region for the measure of synchrony (PLV) in the alpha frequency band. The positive association between participation in the left temporal region and VSTM scores for the Low participation subgroup (regression test; no association with cognition for the High participation older subgroup). **b** Distribution of the young adults and older adults' subgroups for the multiplex participation coefficient in the right parietal region in the alpha frequency band. The positive association between participation in the right parietal region and VSTM scores for the Low participation older subgroup (regression test; no association with cognition for the high participation older subgroup). **c** Distribution of young adults and older adults' subgroups for the multiplex participation coefficient in the right parietal region in the alpha frequency band for the measure of directionality (TE). The positive association between the participation of the right parietal region and VSTM scores for the Low participation subgroup (regression test; negative association with cognition for the High participation older subgroup: $r = -0.491$, $p = 0.033$). The level of education was controlled as a covariate. All results were adjusted for multiple comparisons using FDR corrections at $q < 0.05$. $n = 46$ participants per group. The black vertical line represents the standard error of the mean. *$p < 0.05$; **$p < 0.01$; ***$p < 0.001$.

structural layer (DWI) than in the functional layer (PLV/TE) for the right parietal region (difference between DWI/PLV and DWI/TE layers, $p = 0.001$; see Fig. S4 in Supplementary data). The left temporal region follows this trend as well (difference between DWI/PLV layers, $p = 0.086$; difference between DWI/TE layers, $p = 0.001$).

Interestingly, we examined the contribution of the different layers of connectivity within both older subgroups compared to the younger group for alpha temporal and parietal functional activity (see Fig. S5). We observed that the older subgroup that showed lower cognitive performance (High participation) did show the difference in contribution between the two functional layers (differences between PLV and TE, $p < 0.001$), in contrast to the Low older subgroup that did show better associations with cognitive performance ($p < 0.05$). These results were found only for the left temporal region.

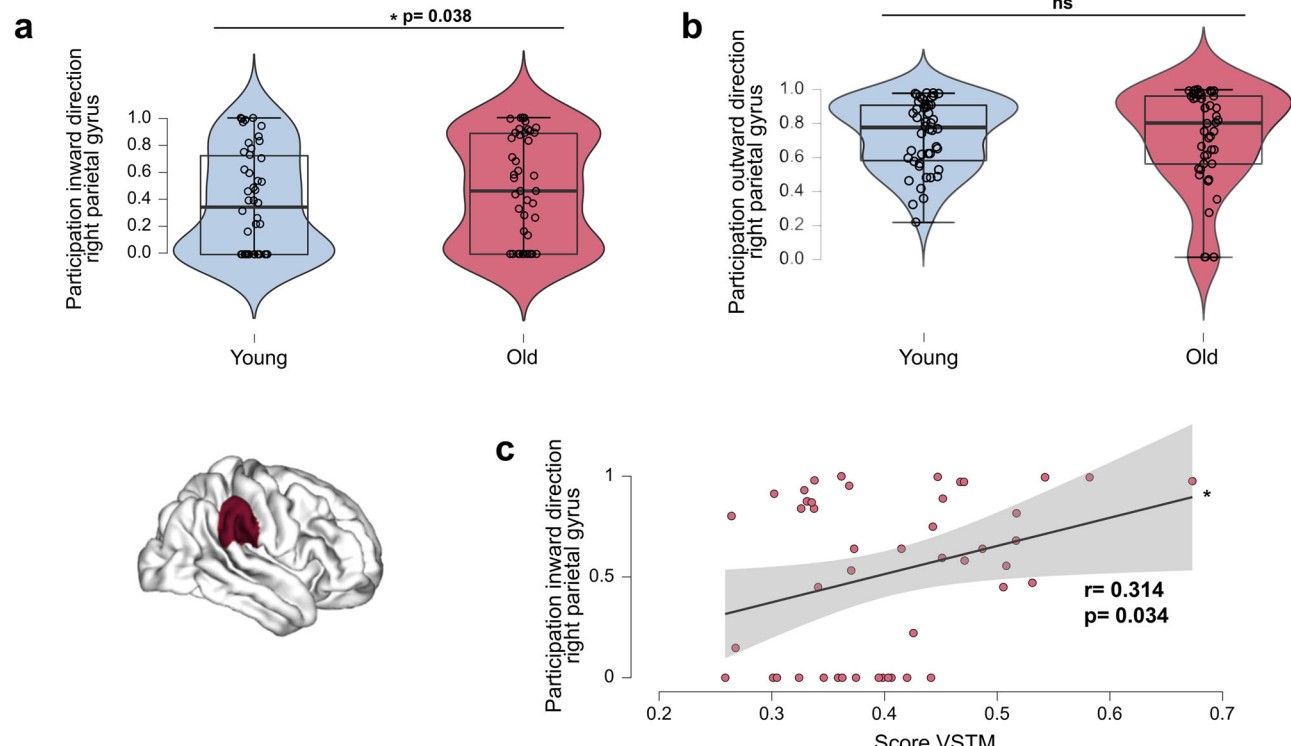

**Fig. 4 | Multiplex participation coefficient level differences between young and older subgroups and association with cognition. a** Increased inward directionality (i.e., directed towards the right parietal region) in older adults relative to younger adults (*t*-test) for the right parietal region in the alpha frequency band. **b** Preserved outward direction (i.e., directed towards other regions of the network) in older adults relative to the younger group for the right parietal region in the alpha frequency band. **c** Positive association between the increased multiplex participation coefficient in the inward direction for the right parietal region in an alpha frequency band and VSTM test scores (regression test) in the older group. The level of education was controlled as a covariate. All results were adjusted for multiple comparisons using FDR corrections at *q* < 0.05. *n* = 46 participants per group. The black vertical line represents the standard error of the mean.*p < 0.05.

### Unique detection of subgroups relative to unimodal network analyses

Finally, we performed unimodal analyses (DWI and MEG) to determine the added value of multiplex analyses relative to functional or structural network investigations (see Fig. S6). Regarding the structural layer, we replicated the significant difference in white matter integrity between young and old groups (*p* < 0.001) on global connectivity data. Regarding the functional layer, we did not find a significant difference between younger and older adults at the global matrix level in the alpha frequency band. At the nodal level, no difference between subgroups was observed in functional or structural networks, in contrast with multilayer analyses.

### Discussion

In this study, we have shown the importance of integrating structural and functional information together to better understand ageing effects. Our objectives were twofold: to investigate changes in the brain structure-function association with age and to determine the impact of changes in this association on cognitive performance in older individuals. Our approach relied on a two-layer multiplex network, with a structural layer based on DWI data and another layer based on resting-state MEG data, to identify changes between younger and older healthy individuals from the Cam-CAN repository and to further understand maintenance[8] and compensation[9] phenomena observed in ageing. Two aspects of functional network connectivity were studied: phase synchrony and directed connectivity. We showed the existence of inter-individual variability at the functional level in older individuals at rest that was associated with cognitive performance. Low structure/function multiplex participation coefficient for structure/synchrony and structure/information transfer in temporal and parietal regions in the alpha frequency band, similar to young adults in the parietal region, was associated with preserved cognitive performance in older individuals.

The multiplex participation coefficient can be considered as an indicator of co-dependence between modalities: a high level of this coefficient would indicate a high similarity of connectivity between brain structure and function, whereas a low coefficient would indicate a dissociation of structure and function connectivity. Subgroup analyses based on this coefficient allowed the detection of heterogeneity within cognitively healthy older individuals. First, we showed that lower levels of structure/synchrony participation relative to younger adults might be beneficial for cognitive performance. Second, using multiplex structure/directed connectivity network analyses, we showed that low levels of participation in the inward direction (i.e., corresponding to couplings directed towards a given region), to a similar level than young adults, for the regions investigated was beneficial for cognitive performance. In contrast, an increase in this coefficient was found to be negatively associated with cognitive performance. These subgroups were not found in unimodal analyses.

The inferior temporal and supramarginal parietal gyri are both considered to be brain structural cores[38]. They are also both part of the default mode network[39,40] (DMN), a network activated at rest and whose activity has been associated with memory and executive performance[41]. Moreover, the alpha frequency band is involved in the structuring of neural rhythms and has notably been associated with attention allocation and the inhibition of couplings not required for the task[42,43]. By assessing the interaction between brain structure and the alpha frequency band, the present results contribute to existing frameworks about this central brain rhythm[42], as they did not consider such an association. Thus, the disengagement of the DMN, as well as the posterior alpha reduction, are critical for cognition and are impacted by aging[44,45]. Results reveal that, at the scale of our study (i.e., early structural damage and small individual differences in microstructural integrity), certain regions and certain frequency bands are more affected than others[46]. Recent work[47] show that structure–function coupling is heterogeneous

according to brain region and frequency band. Structure–function coupling was found to be greater in the slower and intermediate frequency bands than in the faster frequency bands. Moreover, the alpha band is a central frequency band that previously showed significant age-related changes[30]. Alpha oscillations may play a role in the activation or deactivation of the DMN[48], and Jann et al.[49] showed that the BOLD correlates of alpha-band synchronisation in the resting state were localised in brain regions involving the DMN. Age-related structural changes would be central to these changes and would impact brain function. Our results could indicate that in the presence of fine changes in brain architecture, some older individuals will show a lower level of participation coefficient (i.e., a dissociation of connectivity patterns between brain structure and function) than others, which may be due to compensatory functional readjustments involving the alpha frequency band. These changes would enable better cognitive performance than individuals who will not make these functional readjustments, with higher levels of participation coefficient (i.e., a stronger association of connectivity patterns between brain structure and function). Future, longitudinal investigations remain important to further clarify this association.

Our results also reveal that the subgroup of older individuals who showed lower structure/function multiplex participation coefficient and for whom these changes were positively associated with cognitive performances showed no difference in contribution (calculated by measuring connectivity levels in each layer) between the phase synchrony and information transfer layers. Conversely, an increase in the contribution of the phase synchrony layer compared to information transfer was found for the group without association with cognition. These results were only observed in the left inferior temporal region. These results could indicate inefficient connectivity in these individuals (i.e., synchronised couplings with little to no information exchange). The observation of synchronised activity may, therefore, be related to cognitive function but may also be dissociated from it. Thus, considering synchrony in association with information transfer seems important to clarify age-related changes and to distinguish efficient communications from inefficient/maladaptive network couplings. These communications are highly dependent on the integrity of the underlying structural network, and investigating the respective contribution of structure and function through a multiplex network could also allow distinguishing these functional connectivity patterns in pathologies. Indeed, an increase in neuronal synchrony can be observed in neurodegenerative pathologies and has been considered as maladaptive changes (for a review, see[50]). Further investigations of this distinction could lead to the identification of new markers of subsequent decline and progression of neurodegenerative pathologies.

Taken together, these results demonstrate the importance of the relationship between brain structure and function, particularly with advancing age. Thus, in a population of healthy older individuals, alterations in white matter fibres appear to influence the stability of the functional networks they underlie. These functional changes would then influence an individual's cognitive performance. The concept of maintenance (8) would, therefore, be characterised by a relative preservation of white matter fibres, with no changes in cerebral function and a relative preservation of cognitive performance. In that case, the link between brain structure and function would, therefore, remain similar to that of younger individuals. Conversely, cognitive decline would be related to fine alterations of white matter fibres without a reorganisation of functional networks. The relationship between brain structure and function is altered and tends towards a stronger similarity, in contrast to younger individuals. Finally, compensation (9) would occur when white matter fibres are altered, and compensatory functional reorganisation takes place. This reorganisation of functional networks would then enable individuals to maintain their cognitive performance. The link between structure and function is thus different, leading to greater dissimilarity in connectivity patterns.

Several methodological considerations should be discussed regarding the reported results. First, the study of resting-state activity partly limits the direct investigation of the neural bases of cognitive processes, as it might be less directly associated with cognitive functioning than task-related

activity[51]. Second, the use of phrase synchrony measures (phase locking value or phase lag index) could potentially be impacted by volume conduction, although the main results were replicated across methods. Furthermore, transfer entropy measures are not affected by volume conduction[52] and provide converging findings to phase synchrony. In addition, multilayer analyses investigate local levels of similarity in connectivity between structure and function for each region of the atlas. Third, the analysis of layer contributions only showed results for the left inferior temporal region, which does not allow us to generalise our interpretations to the entire brain. Thus, the pattern of layer contributions may be different in other regions and frequency bands[2], although the reported changes were central in the context of healthy ageing. Longitudinal studies could further validate our interpretations and improve our knowledge of other brain regions. Fourth, the reduced size of the subgroups may lead to spurious correlations. Complementary analyses using the participation coefficient as a continuous variable have, however, enabled replication of the results observed in the older subgroup analyses. Fifth, the age groups studied do not allow us to see the changes in the structure/function link that may occur in middle age. However, these comparisons between young and old groups are in line with previous studies on healthy ageing. Further longitudinal studies of this link are needed to complete our results. Sixthly, in addition to the use of measures such as phase synchrony and entropy transfer with MEG, other methods are also relevant for studying the dynamics of brain connectivity, in particular with fMRI[53]. Seventh, this study, like the previous ones, is still based on correlational evidence, particularly in the link between cerebral connectivity and cognition. However, our approach allows us to better consider the complexity of the relationship between structure and function. Eighth, in this study, we used sLoreta to determine source locations. Although alternative source reconstruction methods could have been used, previous work from the team showed consistent results across methods[54]. Finally, these results provide a better understanding of the relationship between brain structure and function, highlighting the influence of fine structural alterations on functional connectivity changes with ageing. Although it seems less consistent with the literature[7], the opposite may also be possible (i.e., an alteration in white matter fibres following a decrease in functional connectivity in certain regions).

Several questions remain about the association between brain structure and function[2,7]. Indeed, this relationship undergoes crucial changes throughout the lifespan, as well as following several pathologies. The structure–function coupling also appears to fluctuate both over time and regionally. Although structural changes appear to drive changes in coupling between regions, brain functions are not solely determined by brain structure. Decreased integrity seems to have an impact on neuronal synchrony and information exchange, and these changes are distinctly associated with cognitive performance in individuals. Although this would be less consistent with age-related changes on structural network[55], this causality can also be reversed, with reduced synchrony and information exchange impacting white matter integrity. Here, we defined multiplex structure-function models in the context of healthy brain ageing to better understand the heterogeneity of these changes across individuals (see Fig. 5 for a schematic representation of this model). In particular, we show its impact on cognitive performance, which improves our knowledge of different theoretical models of ageing, such as concepts of cognitive maintenance[8] and compensation[9]. Maintenance would thus be characterised by an imbalance in the contribution of phase synchrony and transfer information, with a higher level of contribution from PLV than from TE. Moreover, the level of similarity of connectivity between brain structure and function would be very low. The cognitive decline would also be associated with an imbalance in the contribution of phase synchrony and transfer of information. However, in contrast to maintenance, the level of similarity of connectivity between brain structure and function would be very high. Finally, Compensation would be characterised by a balance in the contribution of phase synchrony and transfer information. The level of similarity of connectivity between structure and brain function would be very low, in the same way as in the maintenance concept. Indeed, a dissociation of connectivity patterns

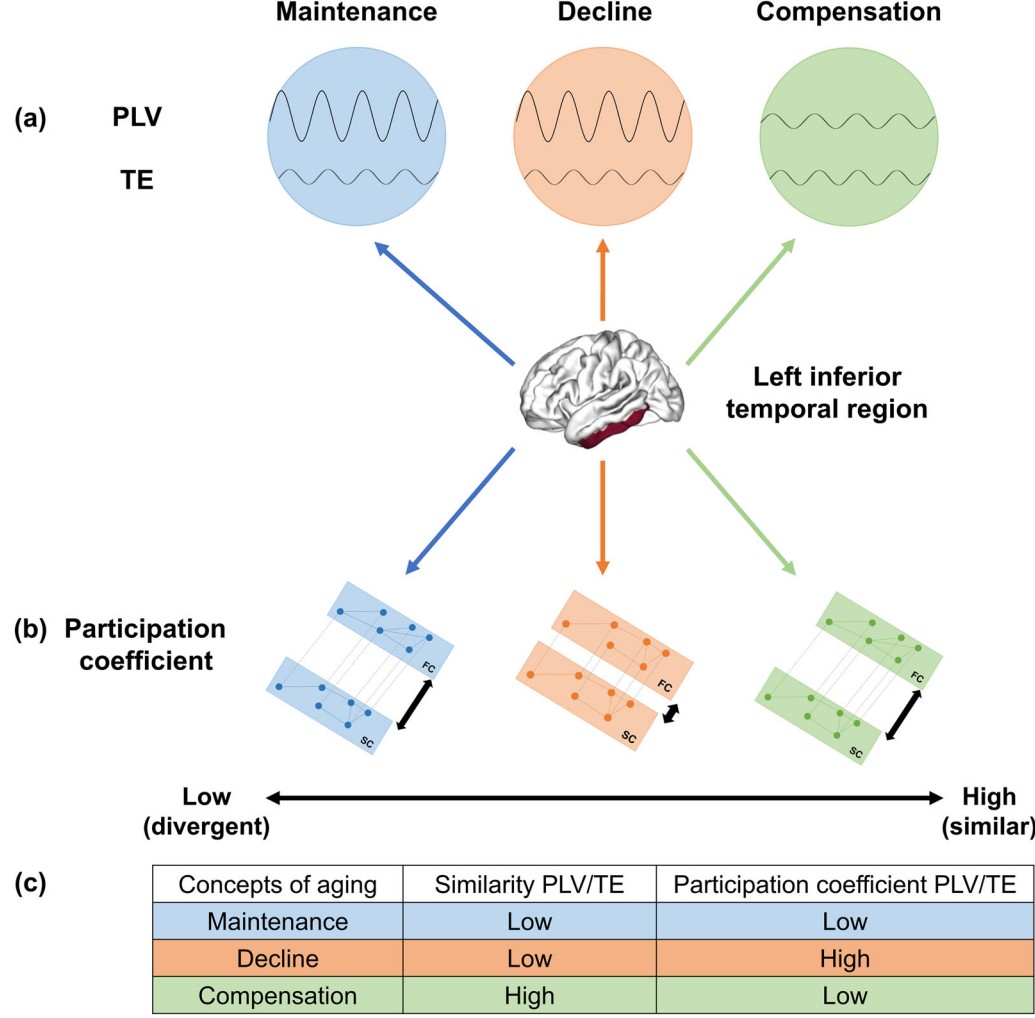

**Fig. 5 | Schematic representation of the proposed model for the left inferior temporal region. a** Level of contribution for PLV and TE. **b** Participation coefficient for PLV/DWI and TE/DWI multiplex network. **c** Summary of the relation between structure and function in the brain. the level of similarity of contribution from PLV/TE, participation coefficient and concepts of ageing. DWI diffusion tensor imaging, PLV phase locking value, TE transfer entropy, FC functional connectivity, SC structural connectivity.

between structure and function has been associated with the preservation of cognitive performance. Importantly, these individual markers were not found in unimodal analyses. This new approach might yield a better understanding of the brain, which could be useful in clinical applications to better understand certain pathologies such as neurodegenerative diseases and, more generally, to further our understanding of the link between structure and function in the brain.

## Methods
### Participants
We selected only individuals in the 20–30 years and 60–70 years age groups who had completed all the neuropsychological tests and brain imaging acquisitions, resulting in 46 individuals per group. All participants aged 20–30 years and 60–70 years were selected from the Cam-CAN database[36,37], in line with the demographic characteristics of individuals recruited in previous works[19,56]. Thus, we analysed data from 46 young (29 women and 17 men; aged 22–29 years) and 46 older healthy adults (29 women and 17 men; aged 60–69 years) whose MEG data have already been published[57] (Table 1). All participants were right-handed, showed normal cognitive functioning[58] (Montreal Cognitive Assessment (MoCA) score >26), and had no neurological or psychiatric conditions. Participants had no depression problems measured with the Hospital Anxiety and Depression Scale (HADS[59]) and self-report (see Fig. S7 in supplementary for a detail of the selection process).

This study is conducted in compliance with the Helsinki Declaration and has been approved by the local ethics committee, Cambridgeshire 2 Research Ethics Committee (reference: 10/H0308/50). Each participant contributed to the study after written informed consent. All ethical regulations relevant to human research participants were followed.

### Behavioural measures
A detailed description of behavioural measures can be found in supplementary materials (see also refs. [36,37]). Cognitive performance was assessed with the mini-mental state evaluation[60] (MMSE) as a measure of general cognitive functioning, the accuracy of the visual short-term memory[61] (VSTM) as a test of short-term memory and working-memory maintenance, the Cattel test[62] measuring reasoning ability, and the Hotel Test[63] assessing executive functions (notably planning abilities). Despite significant differences between the two groups, all participants had normal cognitive functions. These variables were added as covariates in statistical analyses.

### MEG, structural MRI and DWI data acquisition
Resting MEG activity was measured for 10 min, eyes closed (sampling rate: 1 kHz, bandpass filter: 0.03–330 Hz) with a 306-channel MEG system. Participants' 3D-T1 MRI images were acquired on a 32-channel 3 T MRI scanner. The following parameters were used: repetition time = 2250 ms; echo time = 2.99 ms; inversion time = 900 ms; flip angle = 9°; field of

**Table 1 | Demographics and scores for both groups younger and older participants**

| Variables | Young adults | Older adults | *p*-Value | *t*-Value |
|---|---|---|---|---|
| Number of participants | 46 | 46 | 1.000 | – |
| Number of women | 29 | 29 | 1.000 | – |
| Age | 26.5 (2.01) | 64.5 (2.85) | 0.001 | −73.887 |
| Years of education | 22.2 (2.873) | 19.1 (3.262) | 0.001 | 4.774 |
| Hospital Anxiety and Depression Scale (HADS) | 2.07 (0.286) | 2.67 (0.360) | 0.193 | −1.313 |
| MMSE | 29.5 (0.863) | 28.9 (1.173) | 0.013 | 2.531 |
| VSTM | 0.5 (0.088) | 0.4 (0.069) | 0.001 | 3.890 |
| Cattell | 37.8 (3.628) | 30.5 (6.285) | 0.001 | 6.766 |
| Hotel_Num_rows | 4.7 (0.585) | 4.3 (1.008) | 0.018 | 2.420 |
| Hotel_Time | 227.7 (119.796) | 326.9 (194.305) | 0.005 | −2.901 |

*MMSE* mini-mental state evaluation, *VSTM* visual short-term memory, *Hotel_num_rows* corresponding to the number of rows performed by the participant, *Hotel_Time* corresponding to the time used to perform all rows by the participant. Differences between the two groups were calculated using a *t*-test.

view = 256 mm × 240 mm × 192 mm; voxel size = 2 mm isotropic; GRAPPA acceleration factor = 2; acquisition time = 4 min and 32 s. DWI data were obtained with the following parameters: repetition time = 9100 ms; echo time = 104 ms; inversion time = 900 ms; field of view = 192 mm × 192 mm; 66 axial slices; voxel size = 2 mm isotropic; B0 = 0.1000/2000 s/mm²; acquisition time = 10 min and 2 s, readout time 0.0684 (echo spacing = 0.72 ms, EPI factor = 96). See https://camcan-archive.mrc-cbu.cam.ac.uk/dataaccess/ for more information.

## MEG data pre-processing

The Elekta Neuromag MaxFilter 2.2 has been applied to MEG data (temporal signal space separation (tSSS): 0.98 correlation, 10 s window; bad channel correction: ON; motion correction: OFF; 50 Hz + harmonics (mains) notch). Afterwards, artefact rejection, filtering (0.3–100 Hz band-pass), temporal segmentation into epochs, averaging and source estimation were performed using Brainstorm[64]. In addition, physiological artefacts (e.g., blinks and saccades) were identified and removed using spatial space projection of the signal. In order to improve the accuracy of the source reconstruction, the FreeSurfer[65] software was used to generate cortical surfaces and automatically segment them from the cortical structures from each participant's T1-weighted anatomical MRI. The advanced MEG model was obtained from a symmetric boundary element method (BEM model[66]; OpenMEEG[67]), fitted to the spatial positions of each sensor[68]. A cortically constrained sLORETA procedure was applied to estimate the cortical origin of the scalp MEG signals. The estimated sources were then smoothed and projected into standard space (i.e., ICBM152 template) for comparisons between groups and individuals while accounting for differences in anatomy (i.e., grey matter). This procedure was applied for the entire recording duration.

## Connectivity analyses

Phase-locking value analyses[69] (PLV) were used to determine the functional synchrony between regions of interest. PLV estimates the variability of phase differences between two regions over time. If the phase difference varies little, the PLV is close to 1 (this corresponds to high synchronisation between the regions), while the low association of phase difference across regions is indicated by a PLV value close to zero. To ensure PLV results did not reflect volume conduction artefacts, additional control analyses were conducted using phase lag index (weighted PLI analyses) replicated our main subgroups analyses results (see Fig. S2). Because PLV is an undirected measure of functional connectivity and to investigate brain dynamics with

complementary metrics, analyses of transfer entropy (TE) have also been conducted. TE measures how a signal can predict subsequent changes in another signal[70]. It thus provides a directed measure of a coupling's strength. If there is no coupling between regions, then TE is close to 0, while TE is close to 1 if there is a strong coupling between two regions. This method quantifies the flow of information between brain regions, which is why we will refer to it as information transfer for the rest of this article. This information transfer enables us to determine the functional role of a brain region, specifying whether it is a transmitter (i.e., the direction of information flow from this region to other brain regions) or a receiver (i.e., the direction of information flow to this region from other brain regions)[34]. Moreover, this complementary measure is not influenced by volume conduction[52].

PLV and TE were computed using these processes selected in Brainstorm and followed the same processing steps. The range of each frequency band was based on the frequency of the individually observed alpha peak frequency (IAF), measured as the average of peaks detected from both occipitoparietal magnetometers and gradiometers. In line with previous work[71] the following frequency bands were considered: Delta (IAF-8/IAF-6), Theta (IAF-6/IAF-2), Alpha (IAF-2/IAF + 2), Beta (IAF + 2/IAF + 14), Gamma (IAF + 15/IAF + 80). The Hilbert transformation was used for the time-frequency decomposition. The number of cycles per frequency was determined based on IAF (see Table S5). To reduce the dimensionality of the data and to preserve the phase of the time series, the first component of the principal component analysis (PCA) decomposition of the time course of activation in each of the 68 regions of interest (ROI) from the Desikan–Killiany brain atlas[72]. The first component, rather than the average activity, was chosen to reduce signal leakage and volume conduction effects[73]. PLV and TE were computed following these processes in Brainstorm[64].

## DWI data pre-processing

Pre-processing of the diffusion data was performed using ExploreDTI[74] and included the following steps: (a) images were corrected for eddy current distortions and participant motion; (b) a non-linear least squares method was applied for diffusion tensor estimation, and (c) deterministic DWI tractography was applied using the following parameters: uniform resolution of 2 mm, fractional anisotropy (FA) threshold of 0.2 (limit: 1), angle threshold of 45°, and fibre length range of 50–500 mm. The network analysis tools in ExploreDTI were used to quantify the FA value of the fibres connecting the regions of the Desikan atlas to obtain similar matrices to MEG data, using Freesurfer's individual cortical parcellation.

## Multiplex network construction and measures

Using BRAPH[75] software (http://braph.org/), a multiplex network was defined for each subject, with two layers: one "structural" layer with DWI tract FA data and one "functional" layer with PLV or TE MEG data (in this study, a simplification of TE was used to determine whether a region was a receiver or sender). TE analyses were performed on each region and distinguished coupling directed from the network towards a given region (i.e., the inward direction) or from a given region towards the rest of the network (i.e., the outward direction). Frequency bands were analysed separately to investigate their respective associations with the structural layer, in line with previous work showing heterogeneous association across frequency bands[76,77]. In each layer, brain regions from the Desikan–Killiany atlas[78] are represented by nodes connected by edges (see a method's summary in Fig. 1). A binary multiplex matrix was calculated from the individual matrices of DWI and MEG data of each participant. Matrices were binarized according to the minimum density observed for DWI in the older adult group (22%). The density, or number of connections for all matrices, was the same across layers. Auto-correlations between regions were excluded from the analyses.

To evaluate across-layer integration, the multiplex participation coefficient[79] was investigated, allowing the quantification of the connectivity similarity of a node across the different layers. The multiplex participation

coefficient of a node $i$ is defined as[79]: $p_i = \frac{M}{M-1}\left[1 - \sum_{\alpha=1}^{M}\left(\frac{k_i^{[\alpha]}}{o_i}\right)^2\right]$ where

M is the number of layers, $k_i^{[\alpha]}$ the degree of node $i$ at the $\alpha - th$ layer and $o_i$ is the overlapping degree of node $i$, $o_i = \sum_{\alpha} k_i^{[\alpha]}$. This coefficient measures how similar the connectivity patterns are in both layers of the multiplex network. Values range between 0 and 1. In particular, a value of 1 means that the node makes the exact connections in both layers, while a value of 0 means that the node's connections in both layers are different from each other. A large participation value indicates that the node may be central or a hub. To determine which layer is driving the observed results, the degree (i.e., number of connections of each layer of the multiplex network for a given region) was also calculated for each group as: $d^{[\alpha]} = \sum_{j=1}^{N} a_{ij}^{[\alpha]}$; where $a_{ij}^{[\alpha]}$ is the link between node $i$ and $j$ in layer $\alpha$.

## Statistics and reproducibility
To assess differences between age groups in multiplex participation for different brain regions, *t*-tests were applied using the Jamovi software (https://www.jamovi.org/; version 1.6.23). Regression analyses were performed in the older adults' group to assess whether the level of participation coefficient for a region was associated with cognitive performance. Afterwards, participants were grouped according to the level of participation coefficient for each region. Two subgroups were then formed: one corresponding to individuals with a high participation coefficient called "High participant group" and another with a low participation coefficient called "Low participant group". The median individuals (four from each group) were removed from subgroup analyses to reduce median split bias. As a result, each subgroup was composed of 19 individuals. Subgroups were also found in young adults but due to the large variability in young individuals, were considered as a single group. *T*-tests were also performed to determine differences between subgroups. Non-parametric correlations were used when the values were non-continuous (for the MMSE, for example). Level of education, grey matter and total intracranial volume were used as a covariate to control it in the various statistical analyses. Results were FDR corrected for multiple comparisons[80] at each step of the analysis, including *t*-tests between age groups, between different frequency bands and regions, and for regressions with cognition. Original degrees of freedom and corrected *p*-values are reported.

## Reporting summary
Further information on research design is available in the Nature Portfolio Reporting Summary linked to this article.

## Data availability
The datasets analysed in this study are available from the Cambridge Centre for Ageing and Neuroscience (http://www.mrc-cbu.cam.ac.uk/datasets/camcan/). Numerical source data for figures and plots can be found in the Supplementary data file.

## Code availability
The analyses are based on open-source Matlab toolboxes: Brainstorm (https://neuroimage.usc.edu/brainstorm/) for MEG data analysis and BRAPH (http://braph.org/) for graph analysis. Brain region representations were created using the ENIGMA toolbox (https://github.com/MICA-MNI/ENIGMA.git), also an open-source toolbox. All the codes can be found on their respective site. No code has been generated internally.

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

## Acknowledgements

This research did not receive any specific grant from funding agencies in the public, commercial, or not-for-profit sectors. The authors declare no competing interests. We would like to thank the reviewers for their advice. Data collection and sharing for this project were provided by the Cambridge Centre for Ageing and Neuroscience (Cam-CAN). Cam-CAN funding was provided by the UK Biotechnology and Biological Sciences Research Council (grant number BB/H008217/1), together with support from the UK Medical Research Council and the University of Cambridge, UK.

## Author contributions

G.J.: Investigation, Analysis, Writing; M.M.: Methodology, Software, Review; A.C.G.: Methodology, Software, Review; G.V.: Methodology, Software, Review; J.B.P.: Methodology, Software, Review; F.E.: Supervision, Review; T.H.: Conceptualisation, Methodology, Supervision, Review.

## Competing interests

The authors declare no competing interests.
