## [Peer review file · Communications Biology]

Reviewers' comments:

Reviewer #1 (Remarks to the Author):

This manuscript uses an open access dataset ("CamCAN") to investigate interesting and important question about structure-function relationships in aging. I commend the authors for using a relatively novel approach to investigate how MEG-based measures of functional connectivity and DWI structural connectivity contribute to age-related changes in cognitive performance. The approach used here has previously been used to study people with Alzheimer's disease and schizophrenia. The manuscript reports that the similarity of connectivity patterns between brain structure and function in the right parietal and left inferior temporal lobe supported cognitive performance in healthy older adults. While this is a potentially interesting study, I have some questions about the data selection, data analysis, and quality assurance procedures applied as well as the interpretations.

1. The subgroup analysis used here might suffer from a variety of statistical and interpretive issues. First, after performing the median split, only 19 subjects remained in each group. Thus, these subgroup analyses might be underpowered to look at correlations with behavior and spurious correlations might be deemed "significant". Can the authors provide a little more justification for why the subgroup analysis approach was applied? Can the same effects/interpretations be drawn without performing the median split (i.e. using participation index as a continuous variable)?
2. Another statistical problem lies in the interpretation of the subgroup analyses. The authors report that there is a positive correlation with participation and cognitive performance in the "low participation" older adult group and no correlation in the high participation group. However, as pointed out by (Nieuwenhuis et al., 2011) this is not an adequate statistical procedure and instead the interaction (i.e. a statistical test on the difference between the two effects) should be directly tested. See: <https://www.nature.com/articles/nn.2886>
3. A more thorough description of how the ROIs are selected and matched is warranted. If I understand correctly, MEG activity was source localized to the 68 regions from Killiany-Desikan atlas and those same regions were used as seeds for the DWI analysis. On page 14/lines 421-424 the authors state: "To reduce the dimensionality of the data, the first component of the principal component analysis (PCA) decomposition of the time course of activation in each of the 68 regions of interest (ROI) from the Desikan-Killiany brain atlas." After the PCA is conducted, how do these data then get related to the DWI data? More clarity is needed.
4. Related to the point above, edges and nodes are used to calculate the structural connectivity metrics. In my mind, this would mean that at least two nodes are needed per edge. However, the results are described as independent ROIs rather than ROI pairs (i.e. nodes). How does the connectivity get attributed to a single ROI (e.g. inferior temporal lobe) rather than a network or pair of nodes?
5. Perhaps a more thorough presentation of the unimodal data would help the reader understand the steps which are intermediate to the calculation of the multiplex data. The authors should consider a more thorough presentation of the unimodal data as well.
6. I appreciate the attempt made by the authors to tie their imaging metrics to different theories of aging (e.g. maintenance, compensation, etc). However, I did not feel that the explanations provided were thorough enough. If the authors are struggling withk the word limit of this journal, perhaps the authors could focus on the particular mechanism that they think best describes the pattern of data reported here.
7. Data selection/analysis procedures should be better characterized:
7a) According to the CamCAN website, there are 647 subjects with MEG data. How were the subjects

chosen for the current study?

7b) Similarly, the current study seems to only examine a subset of the cognitive data. Why were these particular tests used?

7c) Was quality control performed on DWI, MEG data? 10 min is a long resting state scan and movement could be an issue. Were subjects with high movement or other artifacts excluded?

Reviewer #2 (Remarks to the Author):

Using the Cam-CAN dataset, which includes both young and elderly subjects, Jauny et al. explored the relationship between brain structure (DTI), function (resting MEG), and cognition to gain insights into the mechanisms of healthy aging, including compensatory, maintenance, and decline processes. While the manuscript is well-written and applying a complex method to show structure-function association, I have a major concern: the authors need to provide a clear distinction between hypothesis-driven and exploratory aspects of their analysis, as both approaches are valid but should be clearly stated.

Overall Comments:

A) Interpretation of Relevant Methods:

1) In the introduction, the authors mentioned, "However, for all these studies, the interpretation of these interactions is limited as it is based on correlational evidence, which does not account for the full complexity of such a relationship."

To the best of my knowledge, the application of a complex network model (or transfer entropy) still does not allow for the interpretation of causal associations between brain structure and function (e.g., in terms of timing). Instead, it only indicates directionality, as it is written in the manuscript. The applied statistics in this research still rely on the correlation between multiplex participation coefficients between DTI and MEG. Although the method itself can account for the complexity of the dataset, it still appears to be a correlational and data-driven approach in terms of interpretation.

2) The biological basis of directionality (inward/onward) is unclear to the reader. What does this mean in terms of network neuroscience and brain aging? Is there any supporting evidence from animal studies or proof-of-concept research on this measurement?

B) Dataset and Analysis:

3) The authors conducted statistical analysis using preselected regions based on significant age group differences ("we identified the regions and frequency bands that differed between age groups"). The applied statistics are not clear and well-justified. Regardless of age-related differences in a region, a structure can drive changes in function (or vice versa, such as plasticity or impairment). Therefore, when investigating structure-function associations, defining a region based on age differences could introduce a bias. It would be thus interesting to see how function-structure associations in other regions behave (independent of age or with age as a moderator) and how they relate to cognition. Given the well-established functional and topographical differences between neural oscillations, it seems likely that there could be other age-related differences associated with participation coefficient in regions and frequency bands beyond parietal and temporal regions in the alpha band.

4) The Materials and Methods section is not entirely clear. Some processing, feature extraction, and analysis steps have not been fully explained or appropriately justified. For example, how was TE computed? Did the authors segment the MEG data before applying TE? Did they implement any criteria to determine the embedding dimension?

5) The rationale for excluding participants from the Cam-CAN cohort is unclear, especially considering that Cam-CAN includes more than 300 individuals with MEG, fMRI, and DTI data. Having a reduced sample size could introduce selection bias.

6) Following source localization, the authors extract a single PCA component, but the rationale and method behind this step are not well elucidated.. Further clarification is needed regarding the purpose and application of this step. (i.e., why signal leakage)

7) The authors mentioned that "results were FDR corrected for multiple comparisons at each step of analysis." However, it is not clear how this correction was established, given the large number of regions from the atlas, multiple band frequencies, and different measures.

8) The authors did not seem to have considered the inclusion of variables of no interest when combining fMRI and DWI data in a two-layer multiplex network. This may lead to spurious correlations. Variables such as head motion, total grey matter, and total intracranial volume should be controlled for to ensure the specificity of the findings.

Overall Recommendation: Clarify the Materials and Methods section.

C) Discussion and Interpretation:

8) If the structural layer contributed most, and if the structure affects the large-scale organization of the brain at many levels, it should not be limited to specific frequencies and regions. Could the authors please elaborate on their findings in this context?

9) What do the results mean in terms of behavior-function associations, specifically the correlation in VSTM?

10) While the authors broadly discuss their findings of temporal and posterior regions connecting to the disengagement of DMN during aging, is it possible that some older individuals show a lower level of participation coefficient than others due to the contribution of other regions that are part of DMN (including ACC, VMPFC, or occipital cortex), not necessarily due to compensatory functional readjustments involving the alpha band?

11) While the explanation of the results is comprehensive, it would be beneficial for the authors to briefly describe what is meant by "information transfer."

Minor Comments:

- In this sentence, a reference is needed: "Previous work has mostly focused on characterizing brain structure (i.e., grey matter and white matter), or brain function (i.e., memory, motor function, or

cognitive control)."

- Neither graph theory nor its application in the brain is recent. (also references) Please edit accordingly: „graph theory has more recently been applied across modalities to study the interaction between structure and function, showing strong associations between these dimensions (Bullmore & Sporns, 2009; Honey et al., 2009).“
- For other regions and frequency bands showing differences not associated with cognitive performance, see Figure S1 in the supplementary materials. However, Figure S1 is not clear; for example, what do the values in parentheses represent?
- In Table S1, the p-value is not clear (whether it represents the differences btw old-young or high-low), as the colors are not visible (green). Please indicate also t-values if the t-test applied.
- The statement, "To our knowledge, no study has investigated the changes of structural and functional connectivity with increasing age using a multiplex approach applied on DWI and MEG (or EEG) data," should not be used as an argument for investigating the research question.

Reviewer #3 (Remarks to the Author):

This is an interesting study. The paper is clear and well written. The topic is of interest, from aging perspective but also, for further understanding the brain function/structure couplings. I have only few major/minor points:

- On figure 2, it seems that the difference is driven by few extreme points. Are the results still significant after removing these 4 to 5 extreme points (part B, old for instance)?
- The main advantage of the MEG over other techniques such as fMRI is the temporal resolution, as stated by the authors. However, the study here seems to focus on static connectivity analysis. To show if the brain dynamics is important, or not, in this context, I would recommend testing an approach that take into account the dynamic of brain networks. See for instance Fong Neuroimage 2019 (Dynamic functional connectivity during task performance and rest predicts individual differences in attention across studies) for a simple approach.
- I highly appreciate how authors tested several connectivity measures to ensure consistency of the results. I would also suggest testing the same for the inverse solution. Beamformer is usually used with MEG, however sLoreta is used here. Showing that the results were consistent, or not, among at least two families of inverse solutions will strengthen the paper.
- How authors explain non-significant results when using only MEG?
- The study is mainly a group-level analysis, which is good at global level. However, the subjects are very heterogenous. I am wondering if the authors have looked at the individual-level PC values with age. Is there is any significant trend of the PC with age (regardless the groups)?
- The toolbox/codes used to compute the connectivity measures are missing. This can unfortunately increase the analytical variability (already alarming the community). Authors are strongly requested to share their code on a Github to allow other researchers to replicate/reproduce the results. Ideally, the code should allow to reproduce all the paper's figures. If GUI were used, please provide enough details to reproduce the analysis.
- Technical details about the computation of the connectivity measures are missing. For instance, it is not clear how PLV was computed, what number of cycles per frequency bands? How phase was

estimated (Hilbert, wavelet...)...

Minor:

- Please cite clearly in the results section when p-values are corrected or not for multiple comparisons
- The controls for the different covariables (education level...) should be also clearly mentioned in the results section.

Reviewer #4 (Remarks to the Author):

Thank you for the opportunity to review “Linking structural and functional changes during aging using multilayer brain network analysis” by Jauny and colleagues. The manuscript describes an investigation of structure-function coupling in healthy aging. The authors analyze the Cam-CAN dataset, and unlike most other studies in this domain, focus on neurophysiological functional connectivity estimated using MEG, rather than fMRI. The authors show that there exist specific spatial patterns, focused in parietal and temporal cortex, that predict individual differences in cognitive performance.

Altogether, this is an interesting study of an under-explored topic. The work is rigorous and the conclusions are measured. I have a few suggestions for improvement:

1. The abstract refers to “diffusion tensor imaging” whereas the rest of the paper (correctly) calls it “diffusion weighted imaging”.
2. Why did the authors focus only on young and old participants, effectively dichotomizing age, when the dataset includes data across the lifespan and is particularly well sampled in middle age?
3. The Results immediately go into models that seek to identify relationships between regional structure-function coupling and cognition, skipping an important step: showing us how structure-function coupling varies over the cortex.
4. It wasn't clear to me whether analyses were constrained to alpha connectivity only or whether the authors explored other bands.

CHANGES BROUGHT TO COMMSBIO-23-2853 *Communications Biology*

Code availability:

The analyses are based on open-source Matlab toolboxes: Brainstorm (<https://neuroimage.usc.edu/brainstorm/>) for MEG data analysis, and BRAPH (<http://braph.org/>) for graph analysis. We now provide an additional table as supplementary material, with individual alpha peaks and associated frequency bands (Table S1 in supplemental information) used in our scripts. Brain region representations were created using the ENIGMA toolbox (<https://github.com/MICA-MNI/ENIGMA.git>), also an open-source toolbox. All the codes is openly accessible on their respective site. No code has been generated internally.

Reviewer #1 (Remarks to the Author):

This manuscript uses an open access dataset (“CamCAN”) to investigate interesting and important question about structure-function relationships in aging. I commend the authors for using a relatively novel approach to investigate how MEG-based measures of functional connectivity and DWI structural connectivity contribute to age-related changes in cognitive performance. The approach used here has previously been used to study people with Alzheimer’s disease and schizophrenia. The manuscript reports that the similarity of connectivity patterns between brain structure and function in the right parietal and left inferior temporal lobe supported cognitive performance in healthy older adults. While this is a potentially interesting study, I have some questions about the data selection, data analysis, and quality assurance procedures applied as well as the interpretations.

1. The subgroup analysis used here might suffer from a variety of statistical and interpretive issues. First, after performing the median split, only 19 subjects remained in each group. Thus, these subgroup analyses might be underpowered to look at correlations with behavior and spurious correlations might be deemed “significant”. Can the authors provide a little more justification for why the subgroup analysis approach was applied? Can the same effects/interpretations be drawn without performing the median split (i.e. using participation index as a continuous variable)?

Response. We thank Reviewer#1 for raising this point. The sample size used here for subgroup analysis is consistent with previous work (Hinault et al., 2021; Jauny et al., 2022). Additional regression analyses were conducted using the participation coefficient as a continuous variable in the older age group as a whole. These analyses replicated our results, with a positive association between the participation coefficient of the inferior temporal and supramarginal regions and the MMSE score (respectively, $p=0.044$, $r=0.329$; $p=0.007$, $r=0.393$, see **Figure 1R** below). We have added these elements to the potential limits of the results in the discussion (p.10).

“Fourth, the reduced size of the subgroups may lead to spurious correlations. Additional analyses using the participation coefficient as a continuous variable have, however, replicated the results observed in the older subgroup analyses.”

Figure 1R. Multiplex participation coefficient level and association with cognition in older group. **(A)** Positive association between level of participation coefficient in left inferior gyrus in alpha frequency band and MMSE score in older adults. **(B)** Positive association between level of participation coefficient in right parietal gyrus in alpha frequency band and MMSE score in older adults. Level of education was controlled as a covariate. All results were adjusted for multiple comparisons using FDR corrections at $q < 0.05$.

* $p < 0.05$ ** $p < 0.01$

Reference:

Hinault, T., Mijalkov, M., Pereira, J. B., Volpe, G., Bakke, A., & Courtney, S. M. (2021). Age-related differences in network structure and dynamic synchrony of cognitive control. *NeuroImage*, 236, 118070.

Jauny, G., Eustache, F., & Hinault, T. (2022). Connectivity dynamics and cognitive variability during aging. *Neurobiology of Aging*, 118, 99–105.

2. Another statistical problem lies in the interpretation of the subgroup analyses. The authors report that there is a positive correlation with participation and cognitive

performance in the “low participation” older adult group and no correlation in the high participation group. However, as pointed out by (Nieuwenhuis et al., 2011) this is not an adequate statistical procedure and instead the interaction (i.e. a statistical test on the difference between the two effects) should be directly tested. See: <https://www.nature.com/articles/nn.2886>

Response. We thank Reviewer#1 for raising this point. It was not possible to perform an ANOVA on correlations between participation levels and cognitive performance. However, the presence of a significant correlation at the whole-group level importantly implies a link with the different levels of participation (see answer to previous point and **Figure 1R**), and is consistent with our sub-group analyses. Finally, the two older subgroups did not differ in terms of cognitive performance, but only in terms of the relationship between the participation coefficient and cognitive performance.

3. A more thorough description of how the ROIs are selected and matched is warranted. If I understand correctly, MEG activity was source localized to the 68 regions from Killany-Desikan atlas and those same regions were used as seeds for the DWI analysis. On page 14/lines 421-424 the authors state: “To reduce the dimensionality of the data, the first component of the principal component analysis (PCA) decomposition of the time course of activation in each of the 68 regions of interest (ROI) from the Desikan-Killiany brain atlas.” After the PCA is conducted, how do these data then get related to the DWI data? More clarity is needed.

Response. This is indeed an important point. As we now further clarify, only MEG data were processed using PCA to extract the time courses of each region, which was necessary to perform phase synchrony and transfer entropy analyses. These data were then used to form functional connectivity matrices. The Desikan-Killiany atlas was defined as a common space between modalities (same matrix size, based on individual cortical parcellation). Tractography and the extraction of a structural connectivity matrix based on the calculation of fibers connecting brain regions were performed on DWI data and did not involve PCA. These two matrices were then used to construct the multilayer network.

4. Related to the point above, edges and nodes are used to calculate the structural connectivity metrics. In my mind, this would mean that at least two nodes are needed per edge. However, the results are described as independent ROIs rather than ROI pairs (i.e. nodes). How does the connectivity get attributed to a single ROI (e.g. inferior temporal lobe) rather than a network or pair of nodes?

Response. Indeed, graph analyses enable the investigation of patterns of connections between regions. Here, we had two connectivity matrices, one for MEG data and one for DWI data. Based on general principles of graph theory, each of these matrices was built separately and contained the connectivity values (structural and functional) of each region with the 67 other regions of the Desikan-Killiany atlas. They were then used to form the multilayer network. Based on multiplex metrics (which measure the similarity of nodes and edges across layers for a given region), we were able to determine the multiplex participation for each region. This variable is based on the similarity of the connectivity pattern of each region across layers of the multilayer network (MEG and DWI). This is why we then obtain one value of multilayer

participation for each region of interest, this value depending on the connectivity of each region in the two layers of the newly-formed network.

5. Perhaps a more thorough presentation of the unimodal data would help the reader understand the steps which are intermediate to the calculation of the multiplex data. The authors should consider a more thorough presentation of the unimodal data as well.

Response. We now provide (**Figure 2R** below and **Figure S6** p.11 of the supplemental information) the differences observed between the young and old groups for DWI and MEG data, respectively, as supplementary data.

Figure 2R. Differences observed between young and old groups for DWI and MEG data separately. **(A)** Greater average integrity of white matter fibers in the younger group than in the older group. **(B)** Greater average synchronization in the alpha frequency band in the younger group than in the older group. * $p < 0.05$

6. I appreciate the attempt made by the authors to tie their imaging metrics to different theories of aging (e.g. maintenance, compensation, etc). However, I did not feel that the explanations provided were thorough enough. If the authors are struggling with the word limit of this journal, perhaps the authors could focus on the particular mechanism that they think best describes the pattern of data reported here.

Response. Thank you for this comment. Interpretations have been developed and added to the Discussion section of the article (p.9-10).

“Taken together, these results demonstrate the importance of the relationship between brain structure and function, particularly with advancing age. Thus, in a population of healthy older individuals, alterations in white matter fibers appear to influence the stability of the functional networks they underlie. These functional changes would then influence individual’s cognitive performance. The concept of maintenance (8) would therefore be characterized by a relative preservation of white matter fibers, with no changes in cerebral function and a relative preservation of cognitive performance. In that case, the link between brain structure and function would therefore remain similar to that of younger individuals. Conversely, cognitive decline would be related to fine alterations of white matter fibers without a reorganization of functional networks. The relationship between brain structure and function is altered and tends towards a stronger similarity, in contrast to younger individuals. Finally, compensation (9)

would occur when white matter fibers are altered, and compensatory functional reorganization takes place. This reorganization of functional networks would then enable individuals to maintain their cognitive performance. The link between structure and function is thus different, leading to greater dissimilarity in connectivity patterns.”

7. Data selection/analysis procedures should be better characterized:

7a) According to the CamCAN website, there are 647 subjects with MEG data. How were the subjects chosen for the current study?

Response. Following Reviewer#1 and Reviewer #2 comments, we now provide additional details about the selection of participants in Method (p.13). This database indeed includes a large number of participants. However, not every participant had completed all the neuropsychological tests or brain imaging sessions. Furthermore, we only selected participants in the age ranges corresponding to our previous work in younger and older adults (Hinault et al., 2021; Jauny et al., 2022). Thus, we selected only those individuals in the 20-30 years and 60-70 years age groups who had completed all the neuropsychological tests and brain imaging acquisitions, resulting in 46 individuals per group.

“We selected only individuals in the 20-30 years and 60-70 years age groups who had completed all the neuropsychological tests and brain imaging acquisitions, resulting in 46 individuals per group.”

7b) Similarly, the current study seems to only examine a subset of the cognitive data. Why were these particular tests used?

Response. We now clarify that every cognitive measure of the CamCAN dataset was considered. The Cattell test (Horn & Cattell, 1966) measuring reasoning ability, and the Hotel Test (Shallice & Burgess, 1991) assessing executive functions (notably planning abilities) were included in regression analyses, but no significant associations were found. Only the tests that showed significant association are discussed. These tests are presented in the method section of the manuscript (p.12) and the complete characterization is included as supplemental information.

“A detailed description of behavioural measures can be found in supplementary materials (see also Refs. (36, 37)). Cognitive performance was assessed with the Mini-Mental State Evaluation(55) (MMSE) as a measure of general cognitive functioning, the Visual Short-Term Memory(56) (VSTM) as a test of short-term memory and working-memory maintenance, the Cattell test(57) measuring reasoning ability, and the Hotel Test(58) assessing executive functions (notably planning abilities).”

“Complete characterization of the cognitive tests. These descriptions are based on what can be found on the site <https://www.cam-can.org/> and in Shafto et al. (2014) and Taylor et al. (2017)”.

MMSE (Mini-Mental State Examination): Test comprising a series of 30 questions divided into 6 categories: evaluation of spatial and temporal orientation abilities, learning abilities, attention and calculation abilities, information recall abilities, language and praxis abilities.

VSTM (Visual Short-Term Memory): View (1–4) coloured discs briefly presented on a computer screen, then after a delay, attempt to remember the colour of the disc that was at a cued location, with response indicated by selecting the colour on a colour wheel (touchscreen input).

Cattell: The test contains four subtests with different types of nonverbal "puzzles": series completion, classification, matrices, and conditions. Correct responses are given a score of 1 for a total maximum score of 46.

Hotel task: Perform tasks in role of hotel manager: write customer bills, sort money, proofread advert, sort playing cards, alphabetise list of names. Total time must be allocated equally between tasks; there is not enough time to complete any one task. The number of actions performed (Hotel_num) and the time taken (Hotel_time) are measured.”

References:

J. L. Horn, R. B. Cattell, Refinement and test of the theory of fluid and crystallized general intelligences. *J. Educ. Psychol.* 57, 253–270 (1966).

T. Shallice, P. W. Burgess, Deficits in strategy application following frontal lobe damage in man. *Brain* 114, 727–741 (1991).

7c) Was quality control performed on DWI, MEG data? 10 min is a long resting state scan and movement could be an issue. Were subjects with high movement or other artifacts excluded?

Response. Regarding MEG data, we eliminated artifacts due to eye blinks, heartbeats and any other artifacts that might arise from participant movement or electrode problems. To do this, we used the SSP procedure in Brainstorm. For DWI data, each participant's data was corrected for EPI/susceptibility distortions, using this procedure in ExploreDTI. To correct data, T1 data were used to unwarp the deformations present in our diffusion data. After this correction procedure, each participant's data went through a thorough data quality assessment, allowing us to visualize any remaining artifacts. No subject was excluded because of excessive artifacts.

Reviewer #2 (Remarks to the Author):

Using the Cam-CAN dataset, which includes both young and elderly subjects, Jauny et al. explored the relationship between brain structure (DTI), function (resting MEG), and cognition to gain insights into the mechanisms of healthy aging, including compensatory, maintenance, and decline processes. While the manuscript is well-written and applying a complex method to show structure-function association, I have a major concern: the authors need to provide a clear distinction between hypothesis-driven and exploratory aspects of their analysis, as both approaches are valid but should be clearly stated.

Overall Comments:

A) Interpretation of Relevant Methods:

1) In the introduction, the authors mentioned, "However, for all these studies, the interpretation of these interactions is limited as it is based on correlational evidence, which does not account for the full complexity of such a relationship."

To the best of my knowledge, the application of a complex network model (or transfer entropy) still does not allow for the interpretation of causal associations between brain structure and function(e.g., in terms of timing). Instead, it only indicates directionality, as it is written in the manuscript. The applied statistics in this research still rely on the correlation between multiplex participation coefficients between DTI and MEG. Although the method itself can account for the complexity of the dataset, it still appears to be a correlational and data-driven approach in terms of interpretation.

Response. We agree with Reviewer#2 that our approach remains correlational, notably regarding the link between brain connectivity and cognition. We have nuanced this sentence in the introduction (p.1) and added this limit in the discussion (p.10) of the article.

"However, for all these studies, the interpretation of these interactions is limited as it does not account for the full complexity of such a relationship."

"Seventh, this study, like the previous ones, is still based on correlational evidence, particularly in the link between cerebral connectivity and cognition. However, our approach allows us to better consider the complexity of the relationship between structure and function."

2) The biological basis of directionality (inward/onward) is unclear to the reader. What does this mean in terms of network neuroscience and brain aging? Is there any supporting evidence from animal studies or proof-of-concept research on this measurement?

Response. Directionality analyses enable to determine the functional role of a brain region, clarifying whether it is a transmitter (i.e., the direction of information flows from this region to other brain regions), or a receiver (i.e., the direction of information flows to this region from other brain regions). Recent work using fMRI in humans has shown the presence of a functional asymmetry between brain regions in terms of afferent and efferent information transfer (Xu et al., 2020). Other computer modeling work has also shown a relationship between network topology and information directionality, in particular, by identifying certain brain regions (or nodes) as targets and sources of information (Moon et al., 2015). We have added this information in the Introduction section (p.2).

"In particular, the functional role of the regions could be affected by changes in brain structure. Indeed recent work using fMRI in humans has shown the presence of a functional asymmetry between brain regions in terms of afferent and efferent information transfer (33). Other computer modeling work has also shown a relationship between network topology and information directionality, in particular, by identifying certain brain regions (or nodes) as targets and sources of information (34)."

References:

Xu, N., Doerschuk, P. C., Keilholz, S. D., & Spreng, R. N. (2021). Spatiotemporal functional interactivity among large-scale brain networks. *Neuroimage*, 227, 117628.

Moon, J. Y., Lee, U., Blain-Moraes, S., & Mashour, G. A. (2015). General relationship of global topology, local dynamics, and directionality in large-scale brain networks. *PLoS computational biology*, 11(4), e1004225.

B) Dataset and Analysis:

3) The authors conducted statistical analysis using preselected regions based on significant age group differences ("we identified the regions and frequency bands that differed between age groups"). The applied statistics are not clear and well-justified. Regardless of age-related differences in a region, a structure can drive changes in function(or vice versa, such as plasticity or impairment). Therefore, when investigating structure-function associations, defining a region based on age differences could introduce a bias. It would be thus interesting to see how function-structure associations in other regions behave (independent of age or with age as a moderator) and how they relate to cognition. Given the well-established functional and topographical differences between neural oscillations, it seems likely that there could be other age-related differences associated with participation coefficient in regions and frequency bands beyond parietal and temporal regions in the alpha band.

Response. Following the comments of Reviewer#2, we now provide additional details regarding the statistical analyses (p.15). Based on our research goal, the main contrast in multiplex analyses was the difference between young adults and older adults. The structure-function association in other regions and frequency bands has been investigated in multilayer analyses, but we did not find any significant differences surviving FDR corrections.

"To assess differences between age groups in multiplex participation for different brain regions, t-tests were applied using the Jamovi software (<https://www.jamovi.org/>; version 1.6.23). Regression analyses were performed in the older adults' group to assess whether the level of participation coefficient for a region was associated with cognitive performance. Afterwards, participants were grouped according to the level of participation coefficient for each region. Two subgroups were then formed: one corresponding to individuals with a high participation coefficient called "High participant group" and another with a low participation coefficient called "Low participant group". The median individuals (four from each group) were removed from subgroup analyses to reduce median split bias. As a result, each subgroup was composed of 19 individuals. Subgroups were also found in young adults but due to the large variability in young individuals, were considered as a single group. T-tests were also performed to determine differences between subgroups. Non-parametric correlations were used when the values were non continuous (for the MMSE for example). Level of education, grey matter and total intracranial volume were used as a covariate to control it in the various statistical analyses. Results were FDR corrected for multiple comparisons(74) at each step of analysis, including t-tests between age groups, between different frequency bands and regions, and for regressions with cognition. Original degrees of freedom and corrected p-values are reported."

4) The Materials and Methods section is not entirely clear. Some processing, feature extraction, and analysis steps have not been fully explained or appropriately justified. For

example, how was TE computed? Did the authors segment the MEG data before applying TE? Did they implement any criteria to determine the embedding dimension?

Response. Following Reviewer#2 and Reviewer#3 comments, we now provide additional details about the processing, feature extraction and analysis step in the Method section (p.14).

“TE measures how a signal can predict subsequent changes in another signal(65). It thus provides a directed measure of a coupling’s strength. If there is no coupling between regions, then TE is close to 0, while TE is close to 1 if there is a strong coupling between two regions. This method quantifies the flow of information between brain regions, which is why we will refer to it as information transfer for the rest of this article. This information transfer enables us to determine the functional role of a brain region, specifying whether it is a transmitter (i.e. the direction of information flow from this region to other brain regions) or a receiver (i.e. the direction of information flow to this region from other brain regions) (34). Moreover, this complementary measure is not influenced by volume conduction (50).

PLV and TE were computed using these processes selected in Brainstorm and follow the same processing steps. The range of each frequency band was based on the frequency of the individually observed alpha peak frequency (IAF), measured as the average of peaks detected from both occipitoparietal magnetometers and gradiometers. In line with previous work(66) the following frequency bands were considered: Delta (IAF-8/IAF-6), Theta (IAF-6/IAF-2), Alpha (IAF-2/IAF+2), Beta (IAF+2/IAF+14), Gamma (IAF+15/IAF+80). The Hilbert transformation was used to the time frequency decomposition. The number of cycles per frequency was determined for each individual based on IAF (see Table S1). To reduce the dimensionality of the data and to preserve the phase of the time series, the first component of the principal component analysis (PCA) decomposition of the time course of activation in each of the 68 regions of interest (ROI) from the Desikan-Killiany brain atlas (67). The first component, rather than the average activity, was chosen to reduce signal leakage and volume conduction (68). PLV and TE were computed following these processes in Brainstorm”.

Table S1. Individual alpha peaks frequency (IAF) and associated frequency bands for each participant.

Participants	IAF	Delta_start	Delta_end	Theta_start	Theta_end	Alpha_start	Alpha_end	Beta_start	Beta_end	Gammas_start	Gamma_end
CC110045	9	1	3	3	7	7	11	11	23	24	89
CC110087	11	3	5	5	9	9	13	13	25	26	91
CC110098	9,5	1,5	3,5	3,5	7,5	7,5	11,5	11,5	23,5	24,5	89,5
CC110126	9	1	3	3	7	7	11	11	23	24	89
CC110174	10	2	4	4	8	8	12	12	24	25	90
CC110319	10	2	4	4	8	8	12	12	24	25	90
CC110411	10	2	4	4	8	8	12	12	24	25	90
CC112141	10	2	4	4	8	8	12	12	24	25	90
CC120065	11	3	5	5	9	9	13	13	25	26	91
CC120120	11	3	5	5	9	9	13	13	25	26	91
CC120166	11	3	5	5	9	9	13	13	25	26	91
CC120182	10	2	4	4	8	8	12	12	24	25	90
CC120208	11	3	5	5	9	9	13	13	25	26	91
CC120218	10	2	4	4	8	8	12	12	24	25	90
CC120276	12	4	6	6	10	10	14	14	26	27	92
CC120309	10	2	4	4	8	8	12	12	24	25	90
CC120313	8,5	0,5	2,5	2,5	6,5	6,5	10,5	10,5	22,5	23,5	88,5
CC120319	11	3	5	5	9	9	13	13	25	26	91
CC120469	11,5	3,5	5,5	5,5	9,5	9,5	13,5	13,5	25,5	26,5	91,5
CC120470	8	0	2	2	6	6	10	10	22	23	88
CC120727	8,5	0,5	2,5	2,5	6,5	6,5	10,5	10,5	22,5	23,5	88,5
CC120764	9	1	3	3	7	7	11	11	23	24	89
CC120795	10	2	4	4	8	8	12	12	24	25	90
CC121106	11	3	5	5	9	9	13	13	25	26	91
CC121144	11	3	5	5	9	9	13	13	25	26	91
CC121158	11,5	3,5	5,5	5,5	9,5	9,5	13,5	13,5	25,5	26,5	91,5
CC121200	10,5	2,5	4,5	4,5	8,5	8,5	12,5	12,5	24,5	25,5	90,5
CC121317	12	4	6	6	10	10	14	14	26	27	92
CC121397	8,5	0,5	2,5	2,5	6,5	6,5	10,5	10,5	22,5	23,5	88,5
CC121411	10,5	2,5	4,5	4,5	8,5	8,5	12,5	12,5	24,5	25,5	90,5
CC121428	11	3	5	5	9	9	13	13	25	26	91
CC121479	11	3	5	5	9	9	13	13	25	26	91

CC121795	8	0	2	2	6	6	10	10	22	23	88
CC122172	11	3	5	5	9	9	13	13	25	26	91
CC122405	10	2	4	4	8	8	12	12	24	25	90
CC122620	10	2	4	4	8	8	12	12	24	25	90
CC210250	9,5	1,5	3,5	3,5	7,5	7,5	11,5	11,5	23,5	24,5	89,5
CC210519	10	2	4	4	8	8	12	12	24	25	90
CC212153	10	2	4	4	8	8	12	12	24	25	90
CC220115	10	2	4	4	8	8	12	12	24	25	90
CC220352	10	2	4	4	8	8	12	12	24	25	90
CC220519	10	2	4	4	8	8	12	12	24	25	90
CC220526	7	-1	1	1	5	5	9	9	21	22	87
CC221033	10	2	4	4	8	8	12	12	24	25	90
CC221209	11	3	5	5	9	9	13	13	25	26	91
CC221373	10	2	4	4	8	8	12	12	24	25	90
CC222555	9	1	3	3	7	7	11	11	23	24	89
CC510086	10	2	4	4	8	8	12	12	24	25	90
CC510163	8	0	2	2	6	6	10	10	22	23	88
CC510258	8,5	0,5	2,5	2,5	6,5	6,5	10,5	10,5	22,5	23,5	88,5
CC510342	7,5	-0,5	1,5	1,5	5,5	5,5	9,5	9,5	21,5	22,5	87,5
CC510355	12	4	6	6	10	10	14	14	26	27	92
CC510392	8,5	0,5	2,5	2,5	6,5	6,5	10,5	10,5	22,5	23,5	88,5
CC510395	9	1	3	3	7	7	11	11	23	24	89
CC510415	8	0	2	2	6	6	10	10	22	23	88
CC510433	9	1	3	3	7	7	11	11	23	24	89
CC510486	10	2	4	4	8	8	12	12	24	25	90
CC510548	9	1	3	3	7	7	11	11	23	24	89
CC510551	9	1	3	3	7	7	11	11	23	24	89
CC510648	9	1	3	3	7	7	11	11	23	24	89
CC520002	9	1	3	3	7	7	11	11	23	24	89
CC520011	6	-2	0	0	4	4	8	8	20	21	86
CC520053	9	1	3	3	7	7	11	11	23	24	89
CC520065	8,5	0,5	2,5	2,5	6,5	6,5	10,5	10,5	22,5	23,5	88,5
CC520078	8,5	0,5	2,5	2,5	6,5	6,5	10,5	10,5	22,5	23,5	88,5

CC520097	9	1	3	3	7	7	11	11	23	24	89
CC520134	9	1	3	3	7	7	11	11	23	24	89
CC520136	9	1	3	3	7	7	11	11	23	24	89
CC520147	9	1	3	3	7	7	11	11	23	24	89
CC520200	10	2	4	4	8	8	12	12	24	25	90
CC520215	10	2	4	4	8	8	12	12	24	25	90
CC520239	9	1	3	3	7	7	11	11	23	24	89
CC520254	10	2	4	4	8	8	12	12	24	25	90
CC520279	10	2	4	4	8	8	12	12	24	25	90
CC520377	10	2	4	4	8	8	12	12	24	25	90
CC520390	10	2	4	4	8	8	12	12	24	25	90
CC520391	10	2	4	4	8	8	12	12	24	25	90
CC520395	9	1	3	3	7	7	11	11	23	24	89
CC520424	9	1	3	3	7	7	11	11	23	24	89
CC520503	10	2	4	4	8	8	12	12	24	25	90
CC520552	9	1	3	3	7	7	11	11	23	24	89
CC520585	8	0	2	2	6	6	10	10	22	23	88
CC520607	9	1	3	3	7	7	11	11	23	24	89
CC520624	10	2	4	4	8	8	12	12	24	25	90
CC520745	8,5	0,5	2,5	2,5	6,5	6,5	10,5	10,5	22,5	23,5	88,5
CC520775	10	2	4	4	8	8	12	12	24	25	90
CC520868	8,5	0,5	2,5	2,5	6,5	6,5	10,5	10,5	22,5	23,5	88,5
CC610051	9	1	3	3	7	7	11	11	23	24	89
CC610071	9,5	1,5	3,5	3,5	7,5	7,5	11,5	11,5	23,5	24,5	89,5
CC620073	10	2	4	4	8	8	12	12	24	25	90
CC620259	10	2	4	4	8	8	12	12	24	25	90
CC620262	9,5	1,5	3,5	3,5	7,5	7,5	11,5	11,5	23,5	24,5	89,5
CC620479	9,5	1,5	3,5	3,5	7,5	7,5	11,5	11,5	23,5	24,5	89,5
CC621184	10	2	4	4	8	8	12	12	24	25	90

Table S1. Individual alpha peaks frequency (IAF) and associated frequency bands for each participant.

5) The rationale for excluding participants from the Cam-CAN cohort is unclear, especially considering that Cam-CAN includes more than 300 individuals with MEG, fMRI, and DTI data. Having a reduced sample size could introduce selection bias.

Response. Following Reviewer#1 and Reviewer #2 comments, we give more details about the selection of participants. Indeed, not every participant had completed all the neuropsychological tests or brain imaging. Furthermore, we only selected participants in the age ranges corresponding to our study design (20-30 years) and (60-70 years), in line with previous work on age-related changes (e.g., Hinault et al., 2021; Jauny et al., 2022). Thus, we selected only those individuals in these age groups who had completed all the neuropsychological tests and brain imaging acquisitions, leaving us with 46 individuals per group.

6) Following source localization, the authors extract a single PCA component, but the rationale and method behind this step are not well elucidated. Further clarification is needed regarding the purpose and application of this step. (i.e., why signal leakage)

Response. Following Reviewer#2 comments, we added more details about the purpose and application about the step of PCA.

“The Hilbert transformation was used to the time frequency decomposition. The number of cycles per frequency was determined based on IAF (see Table S1). To reduce the dimensionality of the data and to preserve the phase of the time series, the first component of the principal component analysis (PCA) decomposition of the time course of activation in each of the 68 regions of interest (ROI) from the Desikan-Killiany brain atlas (67). The first component, rather than the average activity, was chosen to reduce signal leakage and volume conduction effects (68).”

References:

(67) Brkić, D., Sommariva, S., Schuler, A. L., Pascarella, A., Belardinelli, P., Isabella, S. L., ... & Pellegrino, G. (2023). The impact of ROI extraction method for MEG connectivity estimation: practical recommendations for the study of resting state data. *NeuroImage*, 120424.

(68) Sato, M., Yamashita, O., Sato, M. A., & Miyawaki, Y. (2018). Information spreading by a combination of MEG source estimation and multivariate pattern classification. *PloS one*, 13(6), e0198806.

7) The authors mentioned that "results were FDR corrected for multiple comparisons at each step of analysis." However, it is not clear how this correction was established, given the large number of regions from the atlas, multiple band frequencies, and different measures.

Response. FDR correction were used for each analysis allowing an a priori selection to reduce the number of comparisons. For each region from the atlas, band frequencies, participation measure and regression with cognition. We have added more information about the FDR correction in method (p.16).

“Results were FDR corrected for multiple comparisons(70) at each step of analysis, including t-tests between age groups, between different frequency bands and regions, and for regressions with cognition.”

8) The authors did not seem to have considered the inclusion of variables of no interest when combining fMRI and DWI data in a two-layer multiplex network. This may lead to spurious correlations. Variables such as head motion, total grey matter, and total intracranial volume should be controlled for to ensure the specificity of the findings.

Response. We thank Reviewer#2 for this comment. We controlled these variables (head motion, total grey matter, and total intracranial volume) and they did not interact with the results presented in this study. These elements are now detailed in the Method section (p.16).

“Level of education, grey matter and total intracranial volume were used as a covariate to control it in the various statistical analyses.”

Overall Recommendation: Clarify the Materials and Methods section.

Response. Following Reviewer#1, Reviewer #2 and Reviewer #3 comments, we clarify The Materials and Method section.

C) Discussion and Interpretation:

8) If the structural layer contributed most, and if the structure affects the large-scale organization of the brain at many levels, it should not be limited to specific frequencies and regions. Could the authors please elaborate on their findings in this context?

Response. Indeed, at the scale of our study, i.e. early structural damage and small individual differences in microstructural integrity, certain regions and certain frequency bands are indeed more affected than others (Mišić et al., 2016). In the study by Liu et al, 2023, the authors show that structure-function coupling is heterogeneous according to brain region and frequency band. Structure-function coupling was found to be greater in the slower and intermediate frequency bands than in the faster frequency bands. In contrast, more massive damage, such as stroke or cranial trauma, affects all brain rhythms and regions. In this study, however, we focus on small changes in white matter integrity. Moreover, alpha is a central frequency band that previously showed significant age-related changes (Courtney & Hinault 2021). This point is now further discussed on p. 11.

“Results reveal that, at the scale of our study (i.e. early structural damage and small individual differences in microstructural integrity), certain regions and certain frequency bands are more affected than others(46). Recent work(47) show that structure-function coupling is heterogeneous according to brain region and frequency band. Structure-function coupling was found to be greater in the slower and intermediate frequency bands than in the faster frequency bands. Moreover, the alpha band is a central frequency band that previously showed significant age-related changes(30).”

References:

- Courtney, S. M., & Hinault, T. (2021). When the time is right: Temporal dynamics of brain activity in healthy aging and dementia. *Progress in neurobiology*, 203, 102076.
- Liu, Z. Q., Shafiei, G., Baillet, S., & Misic, B. (2023). Spatially heterogeneous structure-function coupling in haemodynamic and electromagnetic brain networks. *NeuroImage*, 278, 120276.
- Mišić, B., Betzel, R. F., De Reus, M. A., Van Den Heuvel, M. P., Berman, M. G., McIntosh, A. R., & Sporns, O. (2016). Network-level structure-function relationships in human neocortex. *Cerebral Cortex*, 26(7), 3285-3296.

9) What do the results mean in terms of behavior-function associations, specifically the correlation in VSTM?

Response. VSTM measures working memory, one of the executive functions most affected by ageing. According to our results, individuals who maintain effective brain synchrony through a strong difference in connectivity between structure and function will effectively maintain the relevant information in working memory over time. The difference in connectivity patterns between brain structure and function would reflect a reorganization of functional connectivity, when structural connectivity is impaired. Conversely, individuals with a similar pattern of connectivity between brain structure and function do not show functional connectivity reorganization to compensate for these structural changes, thus impacting the synchronization and maintenance over time of relevant information in working memory. Following suggestions of Reviewer#1 and Reviewer#2 interpretations have been developed and added to the discussion of the article (p.9-10).

“Taken together, these results demonstrate the importance of the relationship between brain structure and function, particularly with advancing age. Thus, in a population of healthy older individuals, alterations in white matter fibers appear to influence the stability of the functional networks they underlie. These functional changes would then influence individual’s cognitive performance. The concept of maintenance (8) would therefore be characterized by a relative preservation of white matter fibers, with no changes in cerebral function and a relative preservation of cognitive performance. In that case, the link between brain structure and function would therefore remain similar to that of younger individuals. Conversely, cognitive decline would be related to fine alterations of white matter fibers without a reorganization of functional networks. The relationship between brain structure and function is be altered and tends towards a stronger similarity, in contrast to younger individuals. Finally, compensation (9) would occur when white matter fibers are altered, and compensatory functional reorganization takes place. This reorganization of functional networks would then enable individuals to maintain their cognitive performance. The link between structure and function is thus different, leading to greater dissimilarity in connectivity patterns.”

10) While the authors broadly discuss their findings of temporal and posterior regions connecting to the disengagement of DMN during aging, is it possible that some older individuals show a lower level of participation coefficient than others due to the contribution of other regions that are part of DMN (including ACC, VMPFC, or occipital

cortex), not necessarily due to compensatory functional readjustments involving the alpha band?

Response. These regions are included in the Desikan-Kiliany atlas used in this study. However, no differences between age groups were observed in these regions.

11) While the explanation of the results is comprehensive, it would be beneficial for the authors to briefly describe what is meant by "information transfer."

Response. We thank Reviewer#2 for this comment and have added more explanation of this concept in the method (p.14).

“This method quantifies the flow of information between brain regions, which is why we will refer to it as information transfer for the rest of this article. This information transfer enables us to determine the functional role of a brain region, specifying whether it is a transmitter (i.e. the direction of information flow from this region to other brain regions) or a receiver (i.e. the direction of information flow to this region from other brain regions) (34).”

Reference:

Moon, J. Y., Lee, U., Blain-Moraes, S., & Mashour, G. A. (2015). General relationship of global topology, local dynamics, and directionality in large-scale brain networks. *PLoS computational biology*, 11(4), e1004225.

Minor Comments:

• In this sentence, a reference is needed: "Previous work has mostly focused on characterizing brain structure (i.e., grey matter and white matter), or brain function(i.e., memory, motor function, or cognitive control)."

Response. Following Reviewer#2 comment, we added a reference in this sentence (p.1).

“Previous work has mostly focused on characterizing brain structure (i.e., grey matter and white matter), or brain function (i.e., memory, motor function or cognitive control)(3).”

Reference:

(3) Sporns, O. (2013). Structure and function of complex brain networks. *Dialogues in clinical neuroscience*.

• Neither graph theory nor its application in the brain is recent. (also references) Please edit accordingly: “graph theory has more recently been applied across modalities to study the interaction between structure and function, showing strong associations between these dimensions (Bullmore & Sporns, 2009; Honey et al., 2009)”.

Response. Following Reviewer#2 comment, we have corrected our statements (p.1).

“However, brain network analysis methods, such as graph theory, have been applied across modalities to study the interaction between structure and function, showing strong associations between these dimensions(5, 6).”

• For other regions and frequency bands showing differences not associated with cognitive performance, see Figure S1 in the supplementary materials. However, Figure S1 is not clear; for example, what do the values in parentheses represent?

Response. Following Reviewer#2 comment, we have added that the values in parentheses correspond to deviations from the mean, in Figure’s legend.

• In Table S1, the p-value is not clear (whether it represents the differences btw old-young or high-low), as the colors are not visible (green). Please indicate also t-values if the t-test applied.

Response. Following Reviewer#2 comment, we have clarified tables S1 and S3 concerning p-values (Underlined: p-value between the younger and the older Low participation groups when older subgroups differ) and added t-values.

Table S1. Demographics and scores for younger group and older subgroups participants formed from multiplex participation coefficient in left temporal region.

Variables	Young adults	Older adults		p-value	t-value
		High participation	Low participation		
				-	-
Number of participants	46	19	19	-	-
Number of women	29	9	10	-	-
Age	26.5 (2.01)	64.4 (2.98)	64.5 (2.84)	< 0.001	-70.442
Years of education	22.2 (2.87)	20.1 (3.3)	19.2 (3.2)	< 0.015	3.659
MMSE	29.5 (0.863)	28.8 (1.4)	29 (1.05)	0.041/0.051	2.388/2.367
VSTM_all	0.5 (0.0875)	0.430 (0.057)	0.430 (0.081)	<0.004	3.985
Cattell	37.8 (3.63)	30.84 (6.89)	30.57 (6.50)	<0.001	6.192
Hotel_Num_rows	4.7 (0.585)	4.21 (0.91)	4.36 (1.16)	0.009/0.108	2.394/1.758
Hotel_Time	227.7 (120)	332.2 (181.9)	334.7 (210.3)	< 0.013	-3.014
Inferiortemporal_participation PLV	0.902 (0.125)	0.977 (0.0217)	0.552 (0.295)	< 0.013	2.792

Figure S1. (A) Table of the five brain regions where multiplex participation coefficient was found to differ between the two groups, young and old (t-test), with the two regions in bold showing significant associations with behavioral performance. **(B)** Visualization of the five brain regions. In blue, regions where multiplex participation coefficient was decreased; in green, regions where multiplex participation coefficient was increased in the older group compared to the younger group. All results were adjusted for multiple comparisons using FDR corrections at $q < 0.05$. Values in parentheses represent deviations from the mean.

Inferiortemporal participation TE	0.379 (0.37)	0.784 (0.25)	0.494 (0.32)	< 0.001/0.244	0.485/-0.127
--	---------------------	---------------------	---------------------	-------------------------	---------------------

The p-values correspond to the highest p-value between the younger group and both older subgroups when older subgroups differ. In bold: p-value statistically different. Underlined: p-value between the younger and the older Low participation groups when older subgroups differ.

VSTM: Visual Short-Term Memory; PLV: Phase Locking Value; TE: Transfer Entropy

Table S2. Demographics and scores for both older subgroups participants formed from multiplex participation coefficient in left temporal region.

Variables	Older adults		p-value	t-value
	High participation	Low participation		
			-	-
Number of participants	19	19	-	-
Number of women	9	10	-	-
Age	64.4 (2.98)	64.5 (2.84)	0.960	0.545
Years of education	20.1 (3.3)	19.2 (3.2)	0.455	-0.522
MMSE	28.8 (1.4)	29 (1.05)	0.799	-0.243
VSTM_all	0.430 (0.057)	0.430 (0.081)	0.992	-0.048
Cattell	30.84 (6.89)	30.57 (6.50)	0.904	-0.447
Hotel_Num_rows	4.21 (0.91)	4.36 (1.16)	0.645	0.375
Hotel_Time	332.2 (181.9)	334.7 (210.3)	0.970	0.062
Inferiortemporal participation PLV	0.977 (0.0217)	0.552 (0.295)	<0.001	-5.768
Inferiortemporal participation TE	0.784 (0.25)	0.494 (0.32)	0.004	-0.771

In bold: p-value statistically different between both older subgroups.

VSTM: Visual Short-Term Memory; PLV: Phase Locking Value; TE: Transfer Entropy

Table S3. Demographics and scores for younger group and older subgroups participants formed from multiplex participation coefficient in right parietal region.

Variables	Young adults	Older adults		p-value	t-value
		High participation	Low participation		
				-	-
Number of participants	46	19	19	-	-
Number of women	29	8	11	-	-
Age	26.5 (2.01)	63.913 (2.85)	65.087 (2.84)	< 0.001	69.899
Years of education	22.2 (2.87)	19.3 (2.81)	19 (2.34)	< 0.001	-4.462
MMSE	29.5 (0.863)	29.31 (0.88)	28.52 (1.42)	0.440 / 0.001	-2.541/ 3.338
VSTM_all	0.5 (0.0875)	0.428 (0.070)	0.443 (0.069)	< 0.014	-3.766
Cattell	37.8 (3.63)	29.78 (5.23)	31.47 (6.89)	< 0.001	-6.668
Hotel_Num_rows	4.7 (0.5852)	4.452 (0.69)	4.21 (1.27)	0.242 / 0.030	-2.060/ 2.279
Hotel_Time	227.7 (119.7)	281.5 (144.14)	347.5 (232.05)	0.129/ 0.009	2.552/ -2.763
Supramarginal participation PLV	0.668 (0.3205)	0.981 (0.016)	0.601 (0.25)	< 0.001 / 0.447	1.626/ 0.960
Supramarginal participation TE	0.351 (0.37)	0.634 (0.36)	0.341 (0.40)	0.007 / 0.923	1.500/ 0.171

The p-values correspond to the highest p-value between the younger group and both older subgroups when older subgroups differ. In bold: p-value statistically different. Underlined: p-value between the younger and the older Low participation groups when older subgroups differ.

VSTM: Visual Short-Term Memory; PLV: Phase Locking Value; TE: Transfer Entropy

Table S4. Demographics and scores for both older subgroups participants formed from multiplex participation coefficient in right parietal region.

Variables	Older adults		p-value	t-value
	High participation	Low participation		
Number of participants	19	19	-	-
Number of women	8	11	-	-
Age	63.913 (2.85)	65.087 (2.84)	0.244	1.349
Years of education	19.3 (2.81)	19 (2.34)	0.718	-0.363
MMSE	29.31 (0.88)	28.52 (1.42)	0.048	-2.048
VSTM_all	0.428 (0.070)	0.443 (0.069)	0.521	0.649
Cattell	29.78 (5.23)	31.47 (6.89)	0.402	0.848
Hotel_Num_rows	4.452 (0.69)	4.21 (1.27)	0.349	-0.949
Hotel_Time	281.5 (144.14)	347.5 (232.05)	0.300	1.052
Supramarginal_participation PLV	0.981 (0.016)	0.601 (0.25)	<0.001	-6.428
Supramarginal_participation TE	0.634 (0.36)	0.341 (0.40)	0.023	-2.372

In bold: p-value statistically different between both older subgroups.

VSTM: Visual Short-Term Memory; PLV: Phase Locking Value; TE: Transfer Entropy

• **The statement, "To our knowledge, no study has investigated the changes of structural and functional connectivity with increasing age using a multiplex approach applied on DWI and MEG (or EEG) data," should not be used as an argument for investigating the research question.**

Response. Following Reviewer#2 comment, we modified this sentence (p.2).

“Thus, multiplex brain networks can be used to study the structure-function relationship in healthy aging. It seems therefore necessary to quantify the heterogeneity of this structure/function relationship in relation to cognitive heterogeneity. Moreover, previous work(30) suggested that alterations in brain structure in presence of delayed and/or noisier brain communications.”

Reviewer #3 (Remarks to the Author):

This is an interesting study. The paper is clear and well written. The topic is of interest, from aging perspective but also, for further understanding the brain function/structure couplings. I have only few major/minor points:

- On figure 2, it seems that the difference is driven by few extreme points. Are the results still significant after removing these 4 to 5 extreme points (part B, old for instance)?

Response. Following Reviewer#3, we have removed the extreme points. Even after removing the extreme points, the t-tests and correlations remain significant (see Figure 3R). We have also noticed an error in this figure, which we have modified.

Figure 3R. Multiplex participation coefficient level differences between young and old groups and association with cognition. **(A)** Distribution of the young and old groups in left inferior temporal region (t-test) for the multiplex participation coefficient in alpha frequency band for the measure of synchrony (PLV) and positive association between this level of multiplex participation coefficient and MMSE score. **(B)** Distribution of the young and old groups in right parietal region (t-test) for the multiplex participation coefficient in alpha frequency band for the measure of synchrony (PLV) and positive association between this level of multiplex participation coefficient and MMSE score, in older adults. Level of education was controlled as a covariate. All results were adjusted for multiple comparisons using FDR corrections at $q < 0.05$. * $p < 0.05$ ** $p < 0.01$

- The main advantage of the MEG over other techniques such as fMRI is the temporal resolution, as stated by the authors. However, the study here seems to focus on static connectivity analysis. To show if the brain dynamics is important, or not, in this context, I would recommend testing an approach that take into account the dynamic of brain networks. See for instance Fong Neuroimage 2019 (Dynamic functional connectivity during task performance and rest predicts individual differences in attention across studies) for a simple approach.

Response. Indeed, one of the main advantages of MEG is its high temporal resolution. The approach used here also enables the study of network dynamics. The functional connectivity matrices of the MEG data used in the multilayer networks were calculated from the variables PLV (Phase Locking Value) and TE (Transfer Entropy) over short periods of time

across the resting-state recording, therefore capturing fine-grain temporal dynamics of brain activity. Nevertheless, this point is now included in the discussion (p.10) of the article.

“Sixthly, in addition to the use of measures such as phase synchrony and entropy transfer with MEG, other methods are also relevant for studying the dynamics of brain connectivity, in particular with fMRI (51).”

Reference:

(51) Fong, A. H. C., Yoo, K., Rosenberg, M. D., Zhang, S., Li, C. S. R., Scheinost, D., ... & Chun, M. M. (2019). Dynamic functional connectivity during task performance and rest predicts individual differences in attention across studies. *NeuroImage*, 188, 14-25.

- I highly appreciate how authors tested several connectivity measures to ensure consistency of the results. I would also suggest testing the same for the inverse solution. Beamformer is usually used with MEG, however sLoreta is used here. Showing that the results were consistent, or not, among at least two families of inverse solutions will strengthen the paper.

Response. The method chosen is consistent with previous studies on the subject (Pascual-Marqui et al., 2018). Moreover, other studies by our team have shown consistent results with different inverse solutions (Hinault et al., 2023). This point is now included in the 'limitations' section of the discussion (p.10) of the article.

“Eighth, in this study, we used sLoreta to determine source location, but the use of Beamformer could also have been consistent in our study. However, the use of sLORETA is in line with previous work.”

References:

Pascual-Marqui, R. D., Faber, P., Kinoshita, T., Kochi, K., Milz, P., Nishida, K., & Yoshimura, M. (2018). Comparing EEG/MEG neuroimaging methods based on localization error, false positive activity, and false positive connectivity. *BioRxiv*, 269753.

Hinault, T., Baillet, S., & Courtney, S. M. (2023). Age-related changes of deep-brain neurophysiological activity. *Cerebral Cortex*, 33(7), 3960-3968.

- How authors explain non-significant results when using only MEG?

Response. We were unable to discriminate the subgroups found using the multilayer approach on the basis of MEG data only. However, we replicate in unimodal analyses (MEG or DWI separately) the global changes observed in multilayers. Results suggest that individual differences are better captured by studying the structure-function association than by differences in brain rhythms alone or in structural integrity alone.

- The study is mainly a group-level analysis, which is good at global level. However, the subjects are very heterogenous. I am wondering if the authors have looked at the individual-level PC values with age. Is there is any significant trend of the PC with age (regardless the groups)?

Response. The regressions enabled us to study the level of participation at group level. Following Reviewer#3 suggestion, we looked at the individual-level PC values with age. We observed a significant effect of age on the individual-level PC, $F(1,9) = 5.99$, $p = 0.016$, $\eta^2 = 0.062$.

- The toolbox/codes used to compute the connectivity measures are missing. This can unfortunately increase the analytical variability (already alarming the community). Authors are strongly requested to share their code on a Github to allow other researchers to replicate/reproduce the results. Ideally, the code should allow to reproduce all the paper's figures. If GUI were used, please provide enough details to reproduce the analysis.

Response. We thank Reviewer#3 for these remarks. The analyses are based on open-source Matlab toolboxes: Brainstorm (<https://neuroimage.usc.edu/brainstorm/>) for MEG data analysis, and BRAPH (<http://braph.org/>) for graph analysis. Brain region representations were created using the ENIGMA toolbox (<https://github.com/MICA-MNI/ENIGMA.git>), also an open-source toolbox. All the codes can be found on their respective site.

- Technical details about the computation of the connectivity measures are missing. For instance, it is not clear how PLV was computed, what number of cycles per frequency bands? How phase was estimated (Hilbert, wavelet...)...

Response. Following Reviewer#2 and Reviewer#3 comments, we added more details about the processing, feature extraction and analysis step in method (p.15).

“PLV and TE were computed using these processes selected in Brainstorm and follow the same processing steps. The range of each frequency band was based on the frequency of the individually observed alpha peak frequency (IAF), measured as the average of peaks detected from both occipitoparietal magnetometers and gradiometers. In line with previous work(62) the following frequency bands were considered: Delta (IAF-8/IAF-6), Theta (IAF-6/IAF-2), Alpha (IAF-2/IAF+2), Beta (IAF+2/IAF+14), Gamma (IAF+15/IAF+80). The Hilbert transformation was used to the time frequency decomposition. The number of cycles per frequency was determined based on IAF (see Table S1). To reduce the dimensionality of the data and to preserve the phase of the time series, the first component of the principal component analysis (PCA) decomposition of the time course of activation in each of the 68 regions of interest (ROI) from the Desikan-Killiany brain atlas. The first component, rather than the average activity, was chosen to reduce signal leakage and volume conduction effects (63). PLV and TE were computed following these processes in Brainstorm.”

Minor:

- Please cite clearly in the results section when p-values are corrected or not for multiple comparisons

Response. Following Reviewer#2 and Reviewer#3 comments, we have added a sentence to the statistics section of the method (p.16), as well as to each results legend.

“Results were FDR corrected for multiple comparisons(70) at each step of analysis. Including t-tests between age groups, between different frequency bands and regions, and for regressions with cognition.”

- The controls for the different covariables (education level...) should be also clearly mentioned in the results section.

Response. Following Reviewer#3 comment, we have added a sentence to the statistics section of the method (p.16), as well as to each results legend

“Level of education, grey matter and total intracranial volume were used as a covariate to control it in the various statistical analyses.”

Reviewer #4 (Remarks to the Author):

Thank you for the opportunity to review “Linking structural and functional changes during aging using multilayer brain network analysis” by Jauny and colleagues. The manuscript describes an investigation of structure-function coupling in healthy aging. The authors analyze the Cam-CAN dataset, and unlike most other studies in this domain, focus on neurophysiological functional connectivity estimated using MEG, rather than fMRI. The authors show that there exist specific spatial patterns, focused in parietal and temporal cortex, that predict individual differences in cognitive performance.

Altogether, this is an interesting study of an under-explored topic. The work is rigorous and the conclusions are measured. I have a few suggestions for improvement:

1. The abstract refers to “diffusion tensor imaging” whereas the rest of the paper (correctly) calls it “diffusion weighted imaging”.

Response. We thank Reviewer#4 for this remark and have corrected the error.

“Here, we used the multilayer brain network analysis on structural (diffusion weighted imaging) and functional (magnetoencephalography) data from the Cam-CAN database.”

2. Why did the authors focus only on young and old participants, effectively dichotomizing age, when the dataset includes data across the lifespan and is particularly well sampled in middle age?

Response. We focus on connectivity changes after 60 because important structural integrity changes were reported at that age (Westlye et al., 2010), and comparison with younger participants are in line with previous work. We agree that the middle age changes are also interesting and this is now listed in the limitation section of the article (p.12).

“Fifth, the age groups studied do not allow us to see the changes in the structure/function link that may occur in middle age. However, these comparisons between young and old groups are in line with previous studies on healthy ageing. Further longitudinal studies of this link are needed to complete our results.”

Reference:

Westlye, L. T., Walhovd, K. B., Dale, A. M., Bjørnerud, A., Due-Tønnessen, P., Engvig, A., ... & Fjell, A. M. (2010). Life-span changes of the human brain white matter: diffusion tensor imaging (DTI) and volumetry. *Cerebral cortex*, 20(9), 2055-2068.

3. The Results immediately go into models that seek to identify relationships between regional structure-function coupling and cognition, skipping an important step: showing us how structure-function coupling varies over the cortex.

Response. Changes in the structure/function link for the other regions of the cortex were studied but showed no significant differences between the young and old groups. We have therefore not included them in our results.

4. It wasn't clear to me whether analyses were constrained to alpha connectivity only or whether the authors explored other bands.

Response. We first investigated age-related differences in delta, theta, alpha, beta and gamma frequency bands (see **Figure S1** in supplementary information for these results). Further analyses were conducted on frequency bands that showed results with cognition. For this reason, only the alpha and gamma frequency bands are presented.

REVIEWERS' COMMENTS:

Reviewer #1 (Remarks to the Author):

The authors have addressed the majority of my previous concerns in their rebuttal. There is one remaining issue I would like the authors to clarify. The authors report that lower participation coefficients were associated with better VSTM performance in older adults (see Figure 3a). Does this mean that lower (as opposed to higher) VSTM scores are "good"? Is this an "error" score as opposed to the more commonly reported "precision" score? I would also suggest clarifying that you are using precision as opposed to Capacity (K) or another metric.

Minor comment: the DTI abbreviation (instead of DWI) is used in the supplement.

Reviewer #2 (Remarks to the Author):

I appreciate the authors for responding to my comments and taking the time to edit the manuscript. (e.g., I enjoyed reading the newer version of the discussion.) However, I still have some minor comments:

Although the authors have indicated their selection of participants, it is still unclear how the participants were chosen from the camcan cohort, e.g., the number of exclusions due to missing data, preprocessing issues, or incomplete data. Therefore, it would be beneficial to include a flow chart detailing the participant selection process, starting from the full participation group ($n > 600$) to the final selected participants ($n = 46 * 2$).

Given the cited reference (Coquelet et al., #56), did the authors consider STAI / STAXI and depression scales, if so, it would be great to indicate these demographic characteristics in the table, as it is indicated in #56 (Coquelet et al.,), because this is what the authors have written: "All participants aged 20-30 years and 60-70 years were selected from the Cam-CAN database(36, 37), in line with demographic characteristics of individuals recruited in previous works(56, 19)."

Regarding the age differences in regions (based on the DK atlas):

In addition to Fig. S1, I would appreciate it if the authors could show t-values of differences projected onto a brain (topography) using the DK atlas (for each frequency band). This would enable readers to understand which brain regions/bands have the highest and lowest t-values, regardless of whether they are statistically significant or not.

Please add the reference for Brainstorm: "PLV and TE were computed following these processes in Brainstorm."

Reviewer #3 (Remarks to the Author):

The authors have satisfactorily addressed all my concerns. I look forward to seeing the final version of this work and believe that it will make a valuable contribution to the field.

Changes brought to COMMSBIO-23-2853A Communications Biology

Reviewer #1 (Remarks to the Author):

The authors have addressed the majority of my previous concerns in their rebuttal. There is one remaining issue I would like the authors to clarify. The authors report that lower participation coefficients were associated with better VSTM performance in older adults (see Figure 3a). Does this mean that lower (as opposed to higher) VSTM scores are "good"? Is this an "error" score as opposed to the more commonly reported "precision" score? I would also suggest clarifying that you are using precision as opposed to Capacity (K) or another metric.

Response. The score used for the VSTM corresponds to the accuracy score. Therefore, a high VSTM score indicates “good” performance for a given individual. The positive correlation between participation coefficient and VSTM performance was only found for the low participation group, meaning that an overall lower similarity between brain structure and function was beneficial for VSTM performance. We have also clarified why the accuracy score was used for the VSTM test in Method section p.15.

“the accuracy of the Visual Short-Term Memory(59) (VSTM) was selected as a test of short-term memory and working-memory maintenance”

Minor comment: the DTI abbreviation (instead of DWI) is used in the supplement.

Response. We thank Reviewer#1 for this remark and have corrected the error in the supplement.

Reviewer #2 (Remarks to the Author):

I appreciate the authors for responding to my comments and taking the time to edit the manuscript. (e.g., I enjoyed reading the newer version of the discussion.) However, I still have some minor comments:

Although the authors have indicated their selection of participants, it is still unclear how the participants were chosen from the camcan cohort, e.g., the number of exclusions due to missing data, preprocessing issues, or incomplete data. Therefore, it would be beneficial to include a flow chart detailing the participant selection process, starting from the full participation group (n>600) to the final selected participants (n=46*2).

Response. Following Reviewer’s 2 comments, we added a flow chart detailing the participant selection process in supplementary data (Figure S7, p.42).

Figure S7. Flow chart detailing the participant selection process.

Given the cited reference (Coquelet et al., #56), did the authors consider STAI / STAXI and depression scales, if so, it would be great to indicate these demographic characteristics in the table, as it is indicated in #56 (Coquelet et al.,), because this is what the authors have written:

“All participants aged 20-30 years and 60-70 years were selected from the Cam-CAN database(36, 37), in line with demographic characteristics of individuals recruited in previous works(56, 19).”

Response. Demographic characteristics correspond here to age, education and scores on various neuropsychological tests. We had no access to STAI/STAXI data in the database. However, we had access to the participants' scores on the Hospital Anxiety and Depression scale (HADS), as well as a self-report of whether they had been diagnosed and treated for anxiety or depression and at what age. None of the participants in this study had problems with

depression. We have added a line to the demographic table indicating the HADS scores. We have also added a sentence on this subject in the methods section, p.15.

Variables	Young adults	Older adults	p-value	t-value
Number of participants	46	46	1.000	-
Number of women	29	29	1.000	-
Age	26.5 (2.01)	64.5 (2.85)	0.001	-73.887
Years of education	22.2 (2.873)	19.1 (3.262)	0.001	4.774
Hospital Anxiety and Depression scale (HADS)	2.07 (0.286)	2.67 (0.360)	0.193	-1.313
MMSE	29.5 (0.863)	28.9 (1.173)	0.013	2.531
VSTM	0.5 (0.088)	0.4 (0.069)	0.001	3.890
Cattell	37.8 (3.628)	30.5 (6.285)	0.001	6.766
Hotel_Num_rows	4.7 (0.585)	4.3 (1.008)	0.018	2.420
Hotel_Time	227.7 (119.796)	326.9 (194.305)	0.005	-2.901

“Participants had no depression problems measured with Hospital Anxiety and Depression scale (HADS; (58)) and self-report.”

Regarding the age differences in regions (based on the DK atlas):

In addition to Fig. S1, I would appreciate it if the authors could show t-values of differences projected onto a brain (topography) using the DK atlas (for each frequency band). This would enable readers to understand which brain regions/bands have the highest and lowest t-values, regardless of whether they are statistically significant or not.

Response. All analyses were performed using matrices extracted from Brainstorm and ExploreDTI software. We are therefore unable to provide a topographical map to visualize these results. However, regions showing significant differences between age groups are highlighted in supplementary data figure S1.

A

Regions	Participation Young group	Participation Old group	p-value	Effect	Frequency bands
Inferiortemporal L	0.901 (0.129)	0.790 (0.277)	≤ 0.041	Smaller participation in older adults	Alpha, Beta, Theta
Rostralmiddlefrontal L	0.840 (0.168)	0.726 (0.336)	≤ 0.042	Smaller participation in older adults	Delta, Gamma
Entorhinal L	0.686 (0.318)	0.821 (0.272)	≤ 0.031	Bigger participation in older adults	Alpha, Delta
Supramarginal R	0.668 (0.343)	0.815 (0.245)	≤ 0.029	Bigger participation in older adults	Alpha, Delta, Theta
Insula R	0.473 (0.383)	0.635 (0.365)	0.040	Bigger participation in older adults	Theta

B
Figure S1. (A) Table of the five brain regions where multiplex participation coefficient was found to differ between the two groups, young and old (t-test), with the two regions in bold showing significant associations with behavioral performance. (B) Visualization of the five brain regions. In blue, regions where multiplex participation coefficient was decreased; in green, regions where multiplex participation coefficient was increased in the older group compared to the younger group. All results were adjusted for multiple comparisons using FDR corrections at $q < 0.05$. Values in parentheses represent deviations from the mean.

Please add the reference for Brainstorm: “PLV and TE were computed following these processes in Brainstorm.”

Response. The reference has been added.

“PLV and TE were computed following these processes in Brainstorm (63)”